# Different Breeding Values Under Uniform Environmental Condition for Milk Production Yield Traits in Holstein-Friesian Cows

**DOI:** 10.3390/ani15010051

**Published:** 2024-12-28

**Authors:** László Bognár, Zsolt Jenő Kőrösi, István Anton, Szabolcs Bene, Ferenc Szabó

**Affiliations:** 1National Association of Hungarian Holstein Friesian Breeders, Lőportár u. 16, H-1134 Budapest, Hungary; 2Institute of Animal Sciences, Georgikon Campus, Hungarian University of Agriculture and Life Sciences, Deák Ferenc u. 16, H-8360 Keszthely, Hungary; 3Department of Animal Sciences, Albert Kázmér Faculty of Agriculture and Food Sciences, Széchenyi István University, Vár t. 2, H-9200 Mosonmagyaróvár, Hungary

**Keywords:** Holstein-Friesian cows, milk, fat and protein production, pedigree BLUP, traditional BLUP, genomically enhanced BLUP

## Abstract

This study compared the phenotypic performance and breeding values (BV) of Holstein-Friesian cows using three BLUP (best linear unbiased prediction) estimation methods: pedigree (BV_Pedigree_), traditional (BV_BLUP_) and genomically enhanced (BV_Genomic_) to validate the different estimation models. The novelty of this study lies in the validation of BV estimation methods under uniform environmental conditions. It also provides valuable insights into the implementation of genomic selection in a real production environment, providing practical guidance for optimizing genetic improvement strategies.

## 1. Introduction

Breeding value (BV) is defined as the genetic potential of a breeding animal as a parent [1]. BV can be estimated from various sources of data on the animal itself, its genetically similar collateral relatives and/or the performance of its offspring. The Best Linear Unbiased Prediction (BLUP) method has been developed to consider performance data from genetically diverse contemporary groups, such as those from different farms. This method is considered unbiased because it incorporates more extensive data in subsequent predictions for the same animal. Various versions of BLUP are now widely used as an advanced statistical model that evaluates all animals in a population.

Subsequently, a genomic BV estimation (BV_Genomic_) method was developed that incorporates genomic data and SNP (single nucleotide polymorphism) information from DNA. This method is based on both the SNP information of an individual and the relationship between SNP information and performance, deregressed from the BLUP data of the reference population [2]. Wiggans et al. [3] discussed the impact of genomic selection on dairy breeding, noting that SNP genotyping has allowed faster genetic progress by reducing the generation interval since young animals or embryos can be genotyped. Two main methods (Bayesian and BLUP) have been extensively studied and applied [4]. Echeverri et al. [5] found that predicting BVs using BLUP, MBLUP and Bayes C in the Holstein-Friesian breed produced different results in terms of the magnitude of the estimated values, although BVs based on animal rankings showed no significant differences. Abaci et al. [6] found that among the different methods, the correlation was highest between BLUP and Bayes Cpi, while the correlation was lowest between BLUP and Bayes A. Despite their complexity, Bayesian methods are less widely used in practical breeding than BLUP 79 methods, as highlighted by Wang et al. [4].

Koivula et al. [7] have compared different estimation methods and found that SNP-BLUP and G-BLUP yielded the same validation reliability, while H-BLUP (H-matrix best linear unbiased prediction) provided slightly higher reliability, indicating a marginal advantage of using H-BLUP for genomic evaluation. Lee et al. [8] observed that the reliability of genomic estimated BVs (GEBVs) increased by an average of 9% over traditional estimated BVs (EBVs), with a 7% increase in cows with test records and about a 4% increase in bulls with progeny records. Herrera et al. [9] have evaluated the accuracy of BV estimations using genomic best linear unbiased prediction (GBLUP) and single-step GBLUP (ssGBLUP) compared to Pedigree BLUP (pBLUP) in Philippine dairy buffaloes, finding that genomic methods provided more accurate predictions than pBLUP. Massender et al. [10] studied two goat breeds and observed a small gain in validation accuracy for GEBV relative to pedigree-based EBV in the Alpine breed but not in the Saanen breed, potentially due to limitations in the validation design. Abdel-Shafy et al. [11] reported that using a genotyping array with 90K markers, the prediction accuracy of 0.61 appears suitable under the current challenges. Zhang et al. [12] compared pedigree BLUP, GBLUP, and ssGBLUP, finding that the reliability of EBV can be enhanced by the use of GEBV.

Hayes et al. [13] have applied GBLUP to a Holstein-Friesian reference population and found that while realized and expected accuracies were reasonably aligned, the expected accuracies did tend to over-predict the realized accuracies by an average of 8% across various traits. Echeverri et al. [5] observed that Spearman correlation coefficients between BVs obtained by different methods exceeded 0.5. The results indicated that while the magnitudes of the BVs using BLUP, Modified BLUP (MBLUP), and Bayes C varied across these methods, the rankings of animals based on their BVs did not differ significantly, and despite methodological differences, the relative genetic merit assigned to each animal remained consistent. Cesarani et al. [14] have reported that the inclusion of genomic information could enhance the accuracy of BV estimations and accelerate genetic progress for milk ability in Italian Simmental cattle. Aguilar et al. [15] evaluated four genetic analysis methods for U.S. Holstein cattle, comparing pedigree-only, combined pedigree-genomic (single-step), full data with pedigree, and multi-step approaches. The single-step method, which integrates both pedigree and genomic information, produced genomic predictions with accuracy and bias similar to multi-step methods and could adapt to various population structures. This approach is expected to become increasingly advantageous as more animals are pre-selected based on genotypes.

Fessenden et al. [16] emphasized the importance of selection indices in breeding programs, particularly for predicting an animal’s genetic potential in terms of economic merit. Their study retrospectively examined the effectiveness of a specific selection index, which included genomically-enhanced predicted transmitting abilities, to determine its capacity to forecast observed lifetime profit in U.S. Holstein cattle. The observation is that selection indices are fundamental in breeding programs, introduced in the 1940s to promote balanced genetic gains across traits influencing productivity and economic outcomes, according to Hazel [17]. These indices combine multiple trait data into a single value, helping to rank animals and guide breeding decisions [18]. Typically, selection indices estimate the genetic potential for overall economic merit [19]. Commercial genomic tests commonly incorporate selection indices, enabling dairy producers to evaluate heifers for strategic culling and breeding [20]. Lourenco et al. [21] found that incorporating genomic data in livestock breeding, particularly in dairy cattle, enhances the accuracy of estimated BVs (EBVs) due to improved insights into relationships and linkage disequilibrium (LD) with quantitative trait loci (QTLs). They noted that while adding female genotypes to a primarily male training set offers minimal accuracy gains for young bull evaluations, genotyping females remains valuable for intra-herd selection and identifying elite dams. Genomic selection thus enables early selection and intensifies genetic progress, particularly in high-selection pathways.

However, genomic BV estimation and selection have been used in cattle breeding for several years, and the permanent control of the applied methods may be beneficial, not only from a scientific but also from a practical point of view.

Although numerous publications exist, studies systematically comparing and evaluating various methods of BV estimation under uniform conditions remain limited, and accessible results derived from a substantial number of cows maintained within the same herd are conspicuously lacking.

The objective of this study was to estimate BVs using three different methods, such as the BV_Pedigree_, BV_BLUP_ and BV_Genomic,_ to compare the most significant productive traits of Holstein-Friesian cows in Hungary. The accuracy, reliability, and usability of the three BV estimation methods were evaluated for first-lactation cows of similar age born and managed within the same herd under uniform conditions.

The novelty of this study lies in the validation of BVs by directly comparing different estimation methods with actual realized performance in uniform environmental conditions, providing new insights into the practical application of genomic selection in Holstein cattle breeding.

## 2. Materials and Methods

This study is based on three types of BV data of 1,616,549 Holstein-Friesian females validated by the phenotypic performance of 190 first lactation progeny cows.

### 2.1. The Sample Population Database and the Estimated Traits

The 1,616,549 females used for BV estimation were kept in different herds on different large-scale dairy farms in Hungary, with an average of 453 milk-recorded, herd-book-registered cows per herd [22]. The housing system employed a loose-housing free-stall barn design, featuring either a common lying area or an open lounging configuration. Milking is usually performed in a milking parlor or on some farms by robotic milking.

The cows were fed a TMR (Total Mixed Ration) based system throughout the year. The ration consisted mainly of maize silage or silage of other cereals and concentrates of cereals and protein sources, supplemented with minerals and vitamins. The proportion of the cows’ daily ration was based on their milk production, lactation or dry period stage.

The 190 first lactation progeny cows used to validate the different BVs were born in the same year between March and September 2018. They were reared in the same herd and kept on the same farm, which is one of the largest commercial Holstein dairies in the country with 1000 milking, milk-recorded cows. This controlled housing and feeding regime was essential for minimizing environmental variation, ensuring that the production traits and genetic evaluations reflected inherent genetic differences.

The 190 cows calved and started their first lactation. After 305 days, complete production and type classification performance data were obtained. The production traits measured were 305-day milk production (MLK, kg), fat production (FAT, kg) and protein production (PRO, kg).

Three types of BV, pedigree (BV_Pedigree_), traditional (BV_BLUP_) and genomically enhanced (BV_Genomic_), were available for all females.

The processing of the performance and BV data was performed according to the method described by Stoop et al. [23].

### 2.2. Breeding Value Estimation Methods

The details of the aforementioned three kinds of BV estimation methods are as below.

The BV_Pedigree_ was calculated as a simple mean value of the BV_BLUP_ of the dam and that of the sire as follows:(1)
BV_Pedigree_ = (BV_BLUPd_ + BV_BLUPs_)/2

(where: BV_Pedigree_ = pedigree BV of the cow; BV_BLUPd_ = BV_BLUP_ of the dam; BV_BLUPs_ = BV_BLUP_ of the sire.)

The BV_BLUP_: Using the BLUP model, two matrices were created. One of these was the database matrix, and the other was the pedigree matrix. The pedigree matrix of relatives included pedigree data for full sibs, half-sibs, sires, dams, and grandparents. BLUP models contained information for maternal genetic effects and maternal permanent environmental effects as random effects. The models were constructed as follows:(2)
y = X_b_ + Z_a_ + W_pe_ + e

(where: “y” is the vector of observations; “b” is the vector of fixed effects; “a” is the vector of random animal effects; “pe” is the random vector of permanent environmental effects; “e” is the vector of random residual effects; X, Z and W are the incidence matrices relating records to fixed, animal and random permanent environmental effects, respectively)

The fixed effects were a herd, year, season, parity and age. Random effects include genetic and environmental influences specific to individual animals but not systematically attributable to the fixed effects.

The BV_Genomic_ was estimated using the following: The SNP effect for BV_Genomic_ was estimated from BV_BLUP_ and its reliability, as well as from the genotype. The derived de-regressed BV (de-regressed proof, DRP) from the BV_BLUP_ was used for the calculation of the direct genomic BV (DGV), according to Van Raden et al. [24].(3)
DRP = PA + (EBV − PA) × (EDC_parents + progeny_/EDC_progeny_)


The DGV is based on the Bayesian multi-QTL model of Meuwissen and Goddard [25], where the effects of dense SNPs across the whole genome are fitted directly without the use of haplotypes or identical-by-descent probabilities [26]. Although the method can be applied for multiple traits simultaneously, the routine genomic evaluations are single trait analyses, i.e., m = 1. For m traits, the model is [23]:(4)$$y_{i}=\mu +u_{i}+\sum_{j=1}^{40,947}z_{ij}q_{j}v_{j}+e_{i}$$
(where: y_i_ = vector of phenotypes (deregressed proofs) of bull i; μ = vector of fixed trait means; u_i_ = vector of random polygenic effects of bull i; q_j_ = vector of random non-scaled SNP effects for SNP j with alleles 0, 1, and 2, where SNP allele 0 corresponds to a missing genotype; v_j_ = random scaling vector for SNP j; z_ij_ = design vector for bull i and SNP _j_— z_ij_ = [0 2 0], [0 1 1], [0 0 2] or [2 0 0] for homozygous (AA), heterozygous (AB), homozygous (BB), or non-genotyped (00) bulls at SNP j, respectively; e_i_ = vector of residuals of bull i.)

### 2.3. Correlation and Regression Analysis

To evaluate the normality of the production trait data, the Kolmogorov–Smirnov test was employed, while Levene’s test was used to assess the homogeneity of variances. A multifactor analysis of variance (ANOVA) for the mentioned traits was conducted. Pearson correlation coefficients were calculated between different BVs (r_g_), moreover between BVs and phenotype (r_gp_), as well as Spearman rank correlation between different BVs (r_s_). Linear regression was employed to evaluate the relationships among phenotypic traits and BVs, as outlined by Turney [27]. Throughout the statistical analysis, the methodologies proposed by Seo et al. [28] were followed. Phenotypic records and EBVs were standardized to z-scores before calculating correlations.

### 2.4. Used Softwares

The data were prepared using Microsoft Excel 2019 and Microsoft Word 2019. The evaluation of the database was carried out using the statistical software package SPSS version 27.0 [29].

## 3. Results

The MLK, FAT and PRO of the studied population were quite favorable, as shown in Table 1. The group of 190 cows was relatively homogeneous in terms of production traits, indicated by the coefficient of variation (CV%) being less than 15%.

Data in Table 2 reveal a significant range between the minimum and maximum values of the different BVs; however, the standard errors (SE) relative to the mean values are relatively low (below 5%). In all instances, the data for BV_Pedigree_ are higher than those for the other two types of BVs. This discrepancy is attributed to the stepwise base change in BV estimation carried out every five years between the parental generation and the offspring cow generation.

Table 3 summarizes the correlation coefficients between BVs and phenotypic performances and that of BVs estimated in three different ways.

The values of correlation coefficients are positive in each case, and most of them are significant. As is seen in the table, the BV_BLUP_ shows a stronger relationship with the phenotypic performances (r_gp_ = 0.61–0.70) than BV_Genomic_ (r_gp_ = 0.31–0.48). The weakest association (r_gp_ = 0.15–0.24) between genotype and phenotype was found in the case of BV_Pedigree_. The BV_Genomic_ showed moderate and strong association (*r*_g_ = 0.66–0.67) with BV_BLUP_. There were also a moderate and strong association (r_g_ = 0.56–0.66) between BV_BLUP_ and BV_Pedigree_ and a little bit lower, moderate or strong relationship (r_g_ = 0.43–0.56) between BV_Genomic_ and BV_Pedigree_.

The rank correlation values (r_g_ = 0.66–0.67) between BV_BLUP_ and BV_Genomic_ indicate a strong (r_s_ = 0.65–0.66) association. The correlation coefficients (r_s_ = 0.12–0.57) between the BV_BLUP_ and the BV_Pedigree_ and between BV_Genomic_ and BV_Pedigree_ also show a moderate association (r_s_ = 0.40–0.56).

The results of regression analyses for BVs against phenotypic traits are presented in Table 4. Significant associations were found at a high level (*p* < 0.01), except for the BV_Pedigree_ for FAT. All regression coefficient (b) values were positive, with the highest observed for BV_BLUP_ (b = 0.17–0.21), followed by BV_Genomic_ (b = 0.10–0.15) and the lowest for BV_Pedigree_ (b = 0.04–0.07). The data clearly demonstrate that the highest determination (R^2^ = 0.37–0.48) for each phenotypic trait was achieved by BV_BLUP_, with BV_Genomic_ showing a lower determination (R^2^ = 0.09–0.23) and BV_Pedigree_ the least (R^2^ = 0.02–0.06). Among the production traits, MLK and FAT were more strongly determined by BVs than PRO. Slope values (b = 0.17–0.21) were positive in each case, being highest for BV_BLUP_, lower for BV_Genomic_ (0.10–0.15), and lowest for BV_Pedigree_ (0.04–0.07).

According to both the correlation and regression results, the BV_BLUP_ method was found to be the most reliable, followed by the BV_Genomic_, and the BV_Pedigree_ method showed the least reliability in the same environmental context.

## 4. Discussion

BV estimation presents significant challenges, such as accurately representing genetic effects, minimizing environmental influences, and enhancing the heritability of specific traits to improve estimation accuracy. Two primary strategies can enhance the precision of predictions: increasing the number of relatives contributing data for evaluation and minimizing environmental effects as much as possible [1].

Large-scale genetic evaluations using BLUP techniques can manage data from multiple herds across a country or even broader geographical areas. Furthermore, genomically enhanced estimation methods facilitate more precise determination of genotypes and allow for the estimation of BVs at earlier ages of the animals. Schefers et al. [30] highlighted that genomic selection enhances genetic progress in dairy cattle by improving the accuracy of genetic predictions for young animals, shortening the generation interval, and increasing selection intensity. These advancements nearly double genetic gains in economically valuable traits. Breeders now routinely use genomically enhanced BVs (GEBVs) to select animals for reproduction, while AI companies employ genomic testing to identify elite bulls and females. This approach has become integral to modern dairy breeding and is expected to remain essential in the industry for future progress. Štrbac et al. [31] evaluated six univariate and two multivariate BLUP models for BV estimation in Holstein cattle, comparing different combinations of the numerator relationship matrix (NRM) and genomic relationship matrix (GRM). Results showed that a combination of NRM and GRM in univariate analysis achieved higher accuracy than NRM alone, while multivariate models with repeated measures gave the highest accuracy for all animals. For genotyped animals, ssGBLUPp provided the most accurate BV estimates. The study suggests transitioning to multivariate, repeated measures until a robust reference population is established.

In our model study, which compared different BV estimation methods, we made efforts to minimize environmental impacts on dairy cow performance.

The results in the same cases are similar; some other cases are slightly different from the data published in the literature.

In this study, positive correlations were consistently observed both between different BVs and between phenotypic performances and BVs. This finding partially aligns with and partially diverges from the results of Echeverri et al. [5], who reported that Spearman correlation coefficients between BVs obtained by different methods exceeded 0.5, whereas linear regression coefficients varied between −2.10 and 1.58.

The correlation coefficients between genotype and phenotype, indicated by BV_BLUP_, suggest a high degree of genetic determination (r_gp_ = 0.61−0.71), as further supported by large values (R^2^ = 0.37−0.48) of the coefficient of determination. BV_Genomic_ demonstrated medium to high genetic determination (r_gp_ = 0.31−0.62, R^2^ = 0.09−0.38), albeit less than that observed with BV_BLUP_. BV_Pedigree_ showed the lowest correlation (r_gp_ = 0.15−0.29) and determination coefficients (R^2^ = 0.02−0.08), indicating a weaker predictive ability for the outcome phenotype compared to the other models.

Considering these correlation and determination coefficients as indicators of accuracy and reliability, the results of our study suggest that BV_BLUP_ is the most accurate and reliable method, followed by BV_Genomic_ and finally by BV_Pedigree_.

Our findings align with those of Echeverri et al. [5] and Abaci et al. [6], who concluded that the prediction of BVs using BLUP, MBLUP and Bayes C displayed differences in terms of magnitude from the estimated values. Similarly, our results are somewhat in line with those of Herrera et al. [9], who reported that genomic methods genomic BLUP and single-step genomic BLUP (GBLUP and ssGBLUP) provide more accurate predictions than BV_Pedigree_, with average accuracies for GBLUP and ssGBLUP at 0.24 and 0.29, respectively, compared to 0.21 and 0.22 for BV_Pedigree_. However, our findings diverge from those of Koivula et al. [7], Lee et al. [8] and Cesarani et al. [14], who observed an increase in the reliability of BV_Genomic_ over BV_BLUP_.

Further, our results slightly differ from those of Zhang et al. [12], who compared BV_Pedigree_ with GBLUP and ssGBLUP, noting that the reliability of estimated BVs could be improved from 0.9% to 3.6%, and the reliability of BV_Genomic_ for the genotyped population could reach 83%.

The variance between our results and those documented in the literature likely stems from the specific conditions of our study’s herd, which was exposed to minimal environmental impacts. This reduced environmental variability might have diminished the observable advantages of BV_Genomic_ estimation methods compared to the BV_BLUP_ approach.

The observation that the BV_Pedigree_ values are consistently higher than the other two types of BVs (BV_BLUP_ and BV_Genomic_) while BV_BLUP_ shows the strongest correlation can be explained by considering the following factors:

Simplified calculation of BV_Pedigree_: BV_Pedigree_ is based solely on the average of parental BVs without accounting for additional genetic or environmental factors. This method does not incorporate phenotypic or genotypic data specific to the individual, leading to an upward bias, especially if the parents’ evaluations are themselves overestimated.

Inclusion of phenotypic data in BV_BLUP_: BV_BLUP_ incorporates phenotypic performance data, adjusting for environmental effects and providing a more individualized estimate of genetic merit. This integration enhances the precision and reliability of the predictions, which likely explains its stronger correlation with actual performance.

Impact of genomic information in BV_Genomic_: BV_Genomic_ uses SNP data to enhance accuracy but depends on the size and structure of the reference population. In cases where the reference population is small or not representative, the genomic prediction may have lower accuracy, potentially leading to weaker correlations compared to BV_BLUP_.

Greater variability in BV_Pedigree_ estimates: Since BV_Pedigree_ lacks adjustments for individual-specific data, it may exhibit greater variability compared to BV_BLUP_ or BV_Genomic_. This could explain why it is consistently higher but less strongly correlated with actual performance.

In uniform environmental conditions, as indicated in the study, BV_BLUP_ may better capture the genetic contribution to traits due to its reliance on phenotypic data, resulting in stronger correlations compared to BV_Pedigree_ or BV_Genomic_.

We hope that by integrating BV results into herd management strategies, producers enhance the overall efficiency and profitability of their operations, demonstrating the practical utility and impact of the genetic evaluation procedures validated in this study.

## 5. Conclusions

This study demonstrated significant differences in the accuracy and reliability of BVs estimated using BV_Pedigree_, BV_BLUP_ and BV_Genomic_ under identical environmental conditions in Hungary. BV_BLUP_ exhibited the highest reliability, followed by BV_Genomic_ and then BV_Pedigree_.

Our finding does not diminish the relevance and applicability of the BV_Genomic_ since it provides critical time-saving advantages, offering earlier predictions for traits such as management, longevity and lifetime performance. It also allows farmers to optimize herd size and reduce maintenance costs. However, BV_BLUP_ may serve as a complementary control, ensuring robustness in genetic evaluations where feasible.

The results obtained in this study highlight the importance of tailoring breeding strategies to specific environmental and population contexts and may improve the accuracy and application of BV estimation methods in dairy production for female selection.

In addition to all of this, in the future, it would be worthwhile to perform the calculations repeatedly on a larger database using mixed mathematical methods (e.g., transformed data). The results obtained on a larger database could provide an opportunity to draw further conclusions.

## Figures and Tables

**Table 1 animals-15-00051-t001:** The basic statistics of the sampled cows.

Trait	MLK (kg)	FAT (kg)	PRO (kg)
Number of cows (n)	190	190	190
Mean value	10,910.50	397.86	365.33
SD	1453.70	45.24	39.80
CV%	13.32	11.37	10.90
SE	105.46	3.28	2.89
Min	6505	260	196
Max	13781	511	451

MLK = 305-day milk production; FAT = 305-day fat production; PRO = 305-day protein production.

**Table 2 animals-15-00051-t002:** Basic statistics of the analyzed breeding values.

Trait/Breeding Value	N	Mean	SE	Min	Max
MLK (kg)					
BV_BLUP_	190	739.16	31.55	−397	1779
BV_Genomic_	190	718.11	32.31	−357	2052
BV_Pedigree_	170	1379.57	29.98	−186	2354
FAT (kg)					
BV_BLUP_	190	34.47	1.00	0	65
BV_Genomic_	190	33.44	1.24	−24	75
BV_Pedigree_	170	55.58	0.97	22	87
PRO (kg)					
BV_BLUP_	190	27.01	0.78	−2	56
BV_Genomic_	190	26.05	0.99	−18	66
BV_Pedigree_	170	49.33	0.91	6	78

MLK = 305-day milk production; FAT = 305-day fat production; PRO = 305-day protein production; BV_BLUP_ = traditional BLUP animal model; BV_Genomic_ = genomically enhanced BLUP animal model BV_Pedigree_ = pedigree BLUP model.

**Table 3 animals-15-00051-t003:** Phenotypic, genetic and rank correlations between breeding values and phenotypic traits.

Correlations (r)	BV_BLUP_	BV_Genomic_	BV_Pedigree_
MILK	r_gp_ = 0.70; *p* < 0.01	r_gp_ = 0.48; *p* < 0.01	r_gp_ = 0.24; *p* < 0.01
BV_BLUP_		r_g_ = 0.67; *p* < 0.01 r_s_ = 0.65; *p* < 0.01	r_g_ = 0.66; *p* < 0.01 r_s_ =0.50; *p* < 0.01
BV_Genomic_			r_g_ = 0.53; *p* < 0.01 r_s_ = 0.40; *p* < 0.01
FAT	r_gp_ = 0.69; *p* < 0.01	r_gp_ = 0.32; *p* < 0.01	r_gp_ = 0.15; NS
BV_BLUP_		r_g_ = 0.67; *p* < 0.01 r_s_ = 0.65; *p* < 0.01	r_g_ = 0.56; *p* < 0.01 r_s_ = 0.57; *p* < 0.01
BV_Genomic_			r_g_ = 0.43; *p* < 0.01 r_s_ = 0.41; *p* < 0.01
PRO	r_gp_ = 0.61; *p* < 0.01	r_gp_ = 0.31; *p* < 0.01	r_gp_ = 0.15; *p* < 0.05
BV_BLUP_		r_g_ = 0.66; *p* < 0.01 r_s_ = 0.66; *p* < 0.01	r_g_ = 0.60; p<0.01 r_s_ = 0.12; NS
BV_Genomic_			r_g_ = 0.56; *p* < 0.01 r_s_ = 0.56; *p* < 0.01

BV_BLUP_ = traditional BLUP animal model; BV_Genomic_ = genomically enhanced BLUP animal model BV_Pedigree_ = pedigree BLUP model; r_gp_ = correlation between the phenotype and genotype; r_g_ = genetic correlation; r_s_ = rank correlation.

**Table 4 animals-15-00051-t004:** Regression analysis results.

Breeding value (Y)	Trait (X)	Slope			Intercept			Fitting	
		b	SE	*p*	a	SE	*p*	R^2^	*p*
BV_BLUP_	MLK	0.21	0.02	<0.01	−1529.56	172.66	<0.01	0.48	<0.01
BV_Genomic_	MLK	0.15	0.02	<0.01	−879.32	216.02	<0.01	0.23	<0.01
BV_Pedigree_	MLK	0.07	0.02	<0.01	665.15	229.32	<0.01	0.06	<0.01
BV_BLUP_	FAT	0.21	0.02	<0.01	−48.80	6.51	<0.01	0.47	<0.01
BV_Genomic_	FAT	0.12	0.03	<0.01	−14.20	10.47	<0.01	0.10	<0.01
BV_Pedigree_	FAT	0.04	0.02	NS	38.83	8.66	<0.01	0.02	NS
BV_BLUP_	PRO	0.17	0.02	<0.01	−33.12	5.75	<0.01	0.37	<0.01
BV_Genomic_	PRO	0.10	0.02	<0.01	−12.12	8.73	<0.01	0.09	<0.01
BV_Pedigree_	PRO	0.05	0.02	<0.05	32.06	8.77	<0.01	0.02	<0.05

MLK = 305-day milk production; FAT = 305-day fat production; PRO = 305-day protein production; BV_BLUP_ = traditional BLUP animal model; BV_Genomic_ = genomically enhanced BLUP animal model BV_Pedigree_ = pedigree BLUP model.

## Data Availability

The original data for this study were collected and generated by the National Association of Hungarian Holstein Friesian Breeders. These data can be reviewed and checked at their facilities. For more information or to request access to the data, please visit their website or contact them directly: Website: www.holstein.hu; Email: bognar@holstein.hu; Postal Address: Lőportár Street 16, H-1134 Budapest, Hungary.

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
