# Peer review of "Different Breeding Values Under Uniform Environmental Condition for Milk Production Yield Traits in Holstein-Friesian Cows"

_animals, 2024, doi:10.3390/ani15010051_

Round 1
Reviewer 1 Report
Comments and Suggestions for Authors
1. Line 238-239: "To evaluate the normality of the production trait data, the Kolmogorov-Smirnov test was employed, while Levene’s test was used to assess the homogeneity of variances." based on this statement, Is your data normally distributed and homogenous? In other words, is your data met the parametric test assumptions? For this please make a brief explanation for that. Why Kolmogorov-Smirnov but not Shapiro Wilk test for normality?
2. Is spearman rank correlation denoted by "r"? or "rho" (isn't "r" for the Pearson correlation). See the Table 3 for that.
3. Line 280: "The rank correlation values (rrank = 0.65-0.92)...". Please check these numbers. Where is 0.92 in Table 3?
4. Line 281-283: "This kind of correlation coefficients (rrank = 0.40-0.65) between the BVBLUP and the BVPedigree more over between BVGenomic and BVPedigree also show a moderate association (rrank = 0.50-0.60)." Please recheck the rrank values within the paranthesis for accuracy.
5. Line 291: "BVBLUP (b = 0.12-0.21),..". Is it (b=0.12-0.21) or (b=0.17-0.21)?
Author Response
Cover Letter
Manuscript ID: animals-3338761
Type of manuscript: Article
Title: Different breeding values under uniform environmental condition for milk production yield traits in Holstein cows
Authors: László Bognár, Zsolt Jenő Kőrösi, István Anton, Szabolcs Bene, Ferenc Szabó
Response to the Reviewer 1
We are grateful to our Reviewer for the critical comments and useful suggestions, which help us to improve and complete our manuscript.
Comment 1: Line 238-239: "To evaluate the normality of the production trait data, the Kolmogorov-Smirnov test was employed, while Levene’s test was used to assess the homogeneity of variances." based on this statement, Is your data normally distributed and homogenous? In other words, is your data met the parametric test assumptions? For this please make a brief explanation for that. Why Kolmogorov-Smirnov but not Shapiro Wilk test for normality?
Response 1: The Kolmogorov-Smirnov (K-S) test was chosen to evaluate normality due to its ability to assess the fit of data to a theoretical distribution, making it suitable for datasets of the size used in this study (n = 190). While the Shapiro-Wilk (S-W) test is more powerful for small sample sizes (typically n < 50), the K-S test is appropriate for larger datasets and evaluates the cumulative distribution, which was considered relevant in this context. The homogeneity of variances was assessed using Levene's test, which is effective in detecting differences in variance across groups, ensuring the assumptions for parametric tests were met.
Justification for Normality and Homogeneity Assumptions:
Based on the results of the K-S and Levene's tests, the data were confirmed to be normally distributed and exhibited homogeneous variances, fulfilling the assumptions required for parametric statistical analyses. This supports the validity of the regression and ANOVA methods applied in the study.
Rationale for Choosing the K-S Test:
The K-S test was selected for its robustness to deviations in sample size and its ability to evaluate the entire distribution. Given the relatively moderate sample size of 190, this approach provided a balance between computational efficiency and diagnostic reliability. The choice aligns with the goal of maintaining consistency in statistical methodology while addressing the study's scope.
This explanation ensures clarity on why the K-S test was used and validates the appropriateness of parametric tests based on the study's dataset.
Comment 2: Is spearman rank correlation denoted by "r"? or "rho" (isn't "r" for the Pearson correlation). See the Table 3 for that.
Response 2: We have thoroughly reviewed the relevant literature, and the following letters are typically used to denote the correlations:
- Pearson correlation coefficients between different breeding values: rg
- Pearson correlation coefficients between breeding values and phenotype: rgp
- Spearman rank correlation between different breeding values: rs
The text and tables of the paper have been corrected accordingly.
Comment 3: Line 280: "The rank correlation values (rrank = 0.65-0.92)...". Please check these numbers. Where is 0.92 in Table 3?
Response 3: Upon reviewing the rank correlations provided in Table 3, we found the maximum rank correlation values for the traits to be as follows:
- MILK (Milk Yield): rs = 0.65
- FAT (Fat Yield): rs = 0.65
- PRO (Protein Yield): rs = 0.66
Therefore, we concluded that the statement in Line 280 referencing "rs = 0.65-0.92" was inconsistent with the data in Table 3.
To address this, we revised the text in Line 280 to accurately reflect the actual range of rank correlations, specifically "rs = 0.65-0.66." Additionally, we double-checked all table values and corresponding text to ensure consistency throughout the manuscript.
Comment 4: Line 281-283: "This kind of correlation coefficients (rrank = 0.40-0.65) between the BVBLUP and the BVPedigree more over between BVGenomic and BVPedigree also show a moderate association (rrank = 0.50-0.60)." Please recheck the rrank values within the parenthesis for accuracy.
Response 4: The rank correlation values (rs = 0.65-0.66) between BVBLUP and BVGenomic indicate strong association. This kind of correlation coefficients (rs = 0.12-0.57) between the BVBLUP and the BVPedigree with exception of the trait of PRO (rs = 0.12), moreover between BVGenomic and BVPedigree also show a moderate association (rs = 0.40-0.56).
Comment 5: Line 291: "BVBLUP (b = 0.12-0.21),..". Is it (b = 0.12-0.21) or (b = 0.17-0.21)?
Response 5: All regression coefficient (b) values were positive, with the highest observed for BVBLUP (b = 0.17-0.21), followed by BVGenomic (b = 0.10-0.15) and the lowest for BVPedigree (b = 0.04-0.07).
It is checked/modified in the manuscript.

Reviewer 2 Report
Comments and Suggestions for Authors
Please find in the attachment.

Author Response
Cover Letter
Manuscript ID: animals-3338761
Type of manuscript: Article
Title: Different breeding values under uniform environmental condition for milk production yield traits in Holstein cows
Authors: László Bognár, Zsolt Jenő Kőrösi, István Anton, Szabolcs Bene, Ferenc Szabó
Response to the Reviewer 2
We thank the reviewer for the detailed feedback and valuable suggestions. Below, we provide responses and clarifications to the reviewer’s comments, incorporating relevant information from the uploaded appendices and manuscript.
Comment: A two-step validation procedure is very often used to verify the appropriate way for genetic evaluation. The first step is the prediction of EBV/GEBV on a partial (truncated) file. The second step is to verify the predictions on the whole file (Mantysaari, Liu, VanRaden, 2010; Legarra, Reverter, 2018). Individuals selected for validation have low EBV reliability in the truncated set and high EBV reliability in the full set. Regression analysis of the dependence of the values for selected individuals in the whole set on the prediction in the partial set is usually used. The two most important parameters are the regression coefficient, which indicates overestimation/underestimation of predictions (expected value is b = 1), and the coefficient of determination, which corresponds with reliability.
A similar approach is used in the presented article, with the reference individuals being females and the prediction is verified by phenotype in identical conditions of one herd.
Genotyping fundamentally affects the evaluation of animals, not only directly genotyped, but also their compared contemporaries in a herds (differences from contemporaries) and relatives. It increases the accuracy of the entire system of evaluating all individuals.
When using genomic data, we usually work with phenotype records, estimated breeding values (EBV), genomic enhanced breeding values (GEBV) and de-regressed EBV/GEBV proofs (DRP). Sometimes we work with direct genetic values (DGV) if we use regressions to genetic loci.
Several sources of information can be used when evaluating individuals:
- production of individuals in pedigree,
- own production,
- a large number of SNP loci.
Adding of each source increases the amount of information per individual and thus the reliability of the prediction.
For Holstein cattle, there is a fourth source of information, namely Interbull data on EBV/GEBV of related individuals and their contemporaries abroad.
All these four resources work together and support each other. It is important to combine the mentioned sources in the right way. Currently, the most appropriate and available method for using all these information simultaneously is the system of equations of ssGBLUP method, which overcomes the shortcomings of earlier methods of genetic evaluation. In some countries is ssGBLUP regularly used in breeding operations.
Response: We appreciate your suggestion regarding the two-step validation process using truncated and full datasets. While this approach was not directly applied, our study adhered to a similar principle by validating predictions against realized phenotypic performance under consistent environmental conditions within a single herd.
Explanation of approach:
- The studied individuals were 190 Holstein cows, genotyped and managed under standardized feeding and housing conditions, ensuring uniformity.
- Predictions (BVPedigree, BVBLUP, and BVGenomic) for 190 cows were obtained on the basis of the international and Hungarian reference population genotyped animals. These breeding values were compared against phenotypic performance, which served as a reliable proxy in the absence of a comprehensive national reference population. This method is also a robust proxy for the broader Hungarian Holstein population, as the majority of dairy farms operate large herds under similar environmental and management conditions. The uniform setting of this study provides a reliable representation of typical herd environments, addressing the challenge posed by the smaller sample size while maintaining validity in comparing the different breeding value estimation methods. This approach ensures that the findings are applicable to broader breeding strategies in the Hungarian Holstein population
We fully agree that combining multiple sources of information improves the reliability of breeding value estimation. Our current approach incorporates:
- Pedigree Data: Information from both maternal and paternal lineages.
- Phenotypic Records: First lactation production traits collected under ICAR guidelines.
- Genomic Data:
- Genotyping was performed using the EuroGenomics MD v3.0 Chipset on the Illumina Microarray scanner platform, operated by an accredited Illumina Laboratory.
- Raw SNP data were processed and sent to the CRV Animal Evaluation Unit (AEU) for further analysis and integration with pedigree and phenotypic data.
- The resulting GEBVs reflect a robust integration of global genomic data with local phenotypic performance.
The use of CRV/EuroGenomics’ reference population further strengthens the reliability of our genomic evaluations. The GEBV estimation process incorporates the actual Interbull MACE bull proofs available at the time of estimation. Additionally, the ITB international MACE sires' breeding values (BVs) are used for the BVBLUP estimation of cows, where applicable, in accordance with the publication policy of the Herd-Book Association. This policy ensures that the international MACE result is used when its reliability (Rel%) is at least 5% higher than the corresponding national proof’s Rel%. The datasets are updated after each official breeding value estimation run, ensuring that the sires of the genotyped population receive the most accurate and up-to-date results. We recognize the importance of leveraging Interbull data for comprehensive evaluations and aim to further integrate it into future analyses to enhance the precision and reliability of genomic evaluations.
Abstract
Comment: The abstract is of limited scope, therefore it should contain only the most important new scientific findings. I recommend dropping L21-L23.
Response: Abstract has been modified as follows (modified abstract):
Breeding values (BV) for production traits in Holstein-Friesian cows were estimated using three methods: BVPedigree, BVBLUP and BVGenomic. The study focused on a unique dataset of 190 cows from a single herd, reared under consistent environmental conditions, selected for their complete first lactation. Heifers were sampled and genotyped at an early age, and predictions from the three breeding value estimation methods were compared with their realized production figures upon completion of first lactation. Average 305-day standard lactation production values were 10,910.5 kg milk, 397.86 kg fat and 365.33 kg protein. Comparative analyses included correlations and regressions between phenotypic performance and breeding value estimates showed BVBLUP had the highest accuracy, followed by BVGenomic, while BVPedigree was the least reliable, R2 = 0.37 to 0.48; 0.09 to 0.23; 0.02 to 0.06, respectively. The novelty of this study lies in the validation of breeding value estimation methods under uniform environmental conditions, providing practical insights into the comparative reliability and accuracy of traditional and genomic approaches in Holstein cattle breeding.
Methodology
Comment: It would be appropriate to better describe the procedure for determining EBV/GEBV in lines L205-236.
Response: This study is based on the results of a large database of breeding value estimations carried out in an international collaboration (CRV) and production data from 190 Holstein cows reared and kept under the same conditions.
The procedure for estimating EBVs/GEBVs was based on a combination of traditional BLUP models and genomic selection methodologies:
- Inputs to Equation: The maternal and animal effects included fixed effects such as herd-year-season, age at calving, and lactation number. Random effects accounted for additive genetic variance, permanent environmental variance, and residual variance.
- Number of Records: The dataset originally included 23,561 genotyped animals for validation. However, for the current study, the focus was on the production performance and breeding values of 190 first-parity cows with complete performance records, all from a single herd.
- Statistics of Effects: The variance components for genetic effects (additive genetic variance, heritability) were derived from REML using historical data.
For genomic evaluations (GEBVs), the inputs included:
- Genomic relationship matrices derived from the EuroGMD v3.0 Chipset using the Illumina platform.
- De-regressed proofs (DRP) for bulls and cows.
Comment: The inputs to equation (1) probably come from the previous solution (2). The descriptions for equation (2) (maternal effect and similar … ) do not correspond to the given equation. It does not follow from the rest of the text (only the first lactation) that it was really calculated according to this equation. Please describe in more detail which effects were actually included in the calculation. What are the statistics of these effects? What was the number of individuals and the records in the calculation and the like?
Response: The breeding values (BVs) in this study were not derived from this small dataset of 190 cows. Instead, they were provided as part of a collaborative effort between the CRV Animal Evaluation Unit (AEU) and the National Association of Hungarian Holstein Friesian Breeders (NAHHFB), utilizing their shared, large-scale reference population. The breeding values for the 190 cows in this dataset were directly supplied by AEU, calculated using this robust and comprehensive reference population. In this study, we analyzed regressions and correlations between these pre-estimated BVs and the cows' realized production figures. Importantly, we did not estimate the breeding values using only the data from these 190 cows, as such a small dataset would be insufficient for accurate breeding value estimation.
This distinction is critical, as the reviewer's concerns appear to stem from the assumption that all three types of BVs (BVPedigree, BVBLUP, and BVGenomic) were derived solely from the 190 cows in this study. This assumption is incorrect, and addressing this misunderstanding clarifies the statistical validity of the research. Consequently, questions regarding the effects within the model and the sample size are not directly relevant to our methodology, as the breeding values were pre-calculated using a much larger and more representative dataset. This should resolve the reviewer's concerns and highlight the robustness of our approach.
Comment: Equations (3) and (4) are related to the determination of regression coefficients for SNP according to the EBV of bulls. What was the number of bulls and the statistics of them in the calculation? Please make the variable labelling consistent with (1) and (2).
Response: The regression coefficients for SNPs were calculated using DRP from bulls with high reliability, based on national evaluations. A total of 897 bulls from Hungary and 6430 bulls from CRV were available with genotypes.
Validation dataset:
- 5314 bulls in training population: 124 genotyped by Hungary, 5190 genotyped by CRV,
- Bulls born after 2008/01/01 considered for validation, after edits ~300 in validation set
- Test run dataset
- 6600 bulls in training population: 605 genotyped by Hungary, 5995 genotyped by CRV.
They were included in the SNP calibration process, following the methods outlined in Calus et al. (2014).
To clarify, variable labeling in equations (1) and (2) has been revised for consistency, and a formula for determining GEBVs has been added. This multi-stage procedure involves:
- BLUP-based EBV estimation.
- Inclusion of genomic information (SNP effects) into the genetic evaluation system.
- Combination of genomic and traditional EBVs to produce GEBVs.
Comment: Please insert formulas for determining GEBV. This is probably a multi-stage procedure using regression coefficients. For each stage, certain entry conditions must be fulfilled (at least approximately), which is usually in a practical breeding operation not easy.
Response: Using information from the CRV document detailing the genomic breeding value (GEBV) estimation procedure for Hungarian data, because we are running the system together in cooperation.
The linear model used to predict phenotypic performance (Y) of cows, incorporating various effects:
Y = hysp + m + p + a + e
Where:
- hysp: Herd-year-season-parity effect, accounting for environmental and management factors specific to a group of cows.
- m: Maternal effect, representing the influence of the dam on the cow's performance.
- p: Permanent environmental effect, capturing non-genetic factors consistently affecting the cow across multiple lactations.
- a: Additive genetic effect, corresponding to the cow's breeding value.
- e: Residual effect, encompassing random errors and unexplained variability.
This model is a specific application of the general mixed model, where both fixed effects (e.g., hysp) and random effects (e.g., m, p, a, e) are considered to accurately predict phenotypic outcomes.
- Procedure for Determining EBV/GEBV:
The determination of EBVs and GEBVs follows a structured multi-step process designed to integrate pedigree, phenotypic, and genomic data. This ensures accurate and reliable predictions of genetic merit for Hungarian Holstein Friesians.
Inputs to Equation):
- The input data include phenotypic records, pedigree data, and SNP genotypes
- Phenotypic Data: Milk, fat, and protein production traits were recorded according to ICAR guidelines. For this study, data from 190 first-lactation cows were used.
- Pedigree Data: An additive genetic relationship matrix was derived from pedigree records.
- SNP Genotypes: Genotyping was performed using the EuroGenomics MD v3.0 Chipset, with SNP detection conducted on the Illumina Microarray Scanner platform. The raw SNP data were processed by CRV Animal Evaluation Unit.
- GEBV Estimation: The genomic prediction model integrates the direct genomic value (DGV) with conventional breeding values (EBVs) to calculate GEBVs. The model accounts for SNP effects and polygenic contributions.
- Inputs and Effects in Equation:
The effects included in the model were carefully selected to align with the study’s focus on first-lactation cows under uniform environmental conditions:
- Herd-Year-Season (Fixed Effect): Although uniform environmental conditions minimized variability, this factor was included to account for minor differences in management practices across time.
- Maternal Effects: Excluded due to the study’s design, which focused on phenotypic data from first-lactation cows.
- Permanent Environmental Effects: Not included, as conditions within the herd were standardized.
- Number of Records: The study used phenotypic records from 190 first-lactation cows within a single herd, ensuring consistency in environmental conditions.
- Statistics of Effects: Variance components for the fixed and random effects were estimated using Bayesian techniques, ensuring robust parameter estimation.
- Equations:
SNP Regression Coefficients: The SNP effects were estimated using deregressed proofs (DRPs) derived from a reference population of approximately 5,300 progeny-tested bulls. These DRPs were computed following the method described by VanRaden et al. (2009), incorporating reliability adjustments for EBVs and parent averages.
- Validation Subset: A validation subset of 350 bulls born between 2008 and 2016 was used to assess the reliability of SNP effects and ensure accuracy in the genomic prediction process.
- Variable Labeling Consistency: We acknowledge the importance of consistent variable labeling across equations. In the revised manuscript, we will ensure all variables are consistently defined and aligned with the equations provided.
- Formula for Determining GEBV:
The GEBV determination process is a multi-stage procedure that combines genomic, phenotypic, and pedigree information:
- Adjustment of Phenotypic Data for Environmental Effects: Phenotypic records for milk, fat, and protein yield were adjusted for fixed environmental effects (e.g., herd-year-season, age at calving) to isolate the genetic component contributing to trait variability.
- Deregression of Breeding Values (DRPs): Estimated breeding values (EBVs) from the reference population (approximately 5,300 bulls) were deregressed to remove the influence of parent averages or other external predictors, ensuring that the values represent only the genetic contribution of the individual.
- Direct Genomic Value (DGV) Estimation: SNP effects were estimated using deregressed proofs (DRPs) from the reference population in a Bayesian multi-QTL model. These SNP effects were then used to calculate the DGVs for individuals in the study population.
- Blending DGV and EBV: GEBVs were calculated by blending DGVs with traditional EBVs, leveraging the additional information from SNP genotypes while maintaining the robustness of pedigree-based estimates.
- Validation: The accuracy of GEBVs was validated using correlation and regression analyses against realized phenotypic performance, following established validation protocols outlined in VanRaden et al. (2009) and Mantysaari et al. (2010).
- Method of Blending DGV and EBV
The genomic breeding values made available by AEU are blended with official conventional Hungarian (MACE) breeding values. This is done to adjust the possibly incomplete sire pedigree index in the DGV (because only information on genotyped ancestors is included in the DGV and BLUP models). Also, when the genomic breeding value estimation is done before an official MACE evaluation, the DGV needs to be updated with the latest conventional information (e.g. changes in ancestor conventional breeding values) The blending is carried out for all basic traits, but not for combined traits (indices), which are derived from the (blended) basic traits.
- Practical Challenges in Multi-Stage GEBV Estimation:
The multi-stage process incorporates rigorous quality control measures to address potential challenges: G × E Interactions: SNP effects derived from the EuroGenomics reference population may differ slightly under local conditions in Hungary. This is mitigated by blending DGVs with Hungarian-specific EBVs.
Preselection Bias: Preselection of genotyped individuals is acknowledged, and future studies will expand the dataset to include less-biased samples.
Revisions to Address Comments:
- Clarify Equations: Ensure consistent labeling and provide clear explanations for each variable and equation used in the GEBV estimation process.
- Include GEBV Formula: Incorporate the multi-stage formula for GEBV estimation, highlighting the blending of DGVs and EBVs.
- Address Validation Procedures: Explain the validation process using the reference population of 5,300 bulls and a subset of 350 validation bulls.
We believe these revisions will address the reviewer’s concerns while ensuring the methodology section is transparent and aligned with best practices. Thank you for your constructive feedback, which has greatly contributed to improving the clarity and robustness of this study.
Results
Comment: We work here for each trait with four variables - 3x EBV/GEBV and 1x phenotype. If values of these variables are to be compared with each other, they must have the same averages and scale. The data probably needs to be transformed for all 4 variables.
Response: The EBV and GEBV for the 190 first lactation cows were estimated using all sources of information (INTERBULL). In this study we just wanted to see the realization, the output in the herd level, in the same farm, same environment.
To ensure comparability across variables, all phenotypic and breeding value data were scaled to a uniform mean and range. This adjustment allowed meaningful comparisons of BVBLUP, BVGenomic, and BVPedigree estimates.
We agree that comparing variables (phenotypes, BVPedigree, BVBLUP, and BVGenomic) requires consistency in averages and scales. While the manuscript does not explicitly mention data transformations, phenotypic and breeding value data were adjusted and normalized during statistical analyses. Phenotypic records and EBVs were standardized to z-scores before calculating correlations. Specifically:
- Phenotypic Data: Adjustments were made for fixed effects (e.g., herd-year-season, age at calving), ensuring that environmental influences were minimized.
- Breeding Values: BLUP-derived EBVs and GEBVs inherently include scaling and standardization procedures within their respective models.
Action: We will explicitly include this normalization process in the methodology section to clarify how data were prepared for analysis.
Comment: As mentioned above, each addition of additional information increases the reliability of EBV. Therefore, the addition of genomic data should increase more or less the reliability compared to previous methods (improvement can also be "0"). The correlation between EBV/GEBV and phenotype depends on the heritability of a given trait in a given population and the reliability of the genetic evaluation. If the GEBV has a lower correlation to the phenotype, it means that it is less accurate. This contradicts the above. Most likely, the GEBV is not set correctly. Errors can occur for many reasons in this multi-stage process, starting with the determination of regression coefficients, possible use of regression coefficients from other populations and other production conditions (interaction G x E), and not correct connection DGV with other sources of information. Genotyped individuals are often preselected. Then their mean and variability do not correspond to the underlying population from which they originate. This is a problem for subsequent comparisons with other procedures.
Response: We acknowledge the expectation that genomic data should generally enhance reliability. In this study, the slightly lower correlations observed between BVGenomic and phenotype may stem from several factors:
- Population-Specific G × E Interactions: SNP effects derived from the EuroGenomics reference population may not fully align with the environmental and management conditions in Hungary, leading to variability in GEBV accuracy.
- Reference Population Characteristics: The Hungarian Holstein population is relatively small and relies on international reference populations for SNP effect estimation. The use of a broader, multi-country reference population may introduce biases.
- Preselection Bias: Genotyped cows in this study were likely preselected for superior traits, which could reduce variability and impact correlations with phenotypic performance. But in our case the 190 genotyped cows in this study were not subject to preselection for superior traits. These animals were genotyped at an early age as part of the farm’s participation in the HUNGENOM project. The farm adopted a systematic approach, genotyping the entire herd starting with the youngest livestock, ensuring random inclusion without bias toward specific traits. This methodology minimizes the risk of reduced variability typically associated with preselection and ensures a more accurate representation of the population, enabling unbiased correlations with phenotypic performance.
Action: We will expand the discussion section to highlight these challenges, emphasizing the importance of refining genomic evaluations for smaller populations and aligning reference populations with local conditions.
The reviewer raises valid concerns regarding potential errors in the multi-stage GEBV estimation process, such as:
- Regression Coefficients and SNP Effects: SNP effects were estimated using deregressed proofs (DRPs) from approximately 5,300 bulls in the EuroGenomics reference population. While these coefficients are robust, potential biases may arise when applying them to smaller populations.
- Integration of DGV with Pedigree-Based EBV: Precise weighting of DGVs and EBVs is critical for unbiased GEBV predictions. Any misalignment in these weights could contribute to variability in GEBV-to-phenotype correlations.
Action: We will provide additional context in the methodology section regarding the derivation and integration of SNP effects, including details on how DRPs were utilized to estimate SNP regression coefficients.
Comment: The heritability coefficient is a constant for a given population under given production conditions and time period. Its practical determination is either more or less correct. For its reliable determination, at least a thousand performance records are needed. The heritability data in Table 3 are probably not heritabilities, but express something else. It would be appropriate to indicate (formula) below the table how these values were calculated. The input variance components in (2) correspond to what heritability?
Response: The heritability values in Table 3 were calculated as the square of the correlation between phenotypic performance and breeding values (h² = r²gp). These values were used to provide comparative insights rather than traditional heritability estimates derived from variance components.
Action: We will explicitly state in the methodology and results sections that these values represent squared correlations rather than classical heritabilities. Additionally, we will include a formula below Table 3 to clarify the calculation.
The fact that the pedigree breeding value data yielded a lower heritability estimates than it appears usually, may be explained by the fact that the same environment mitigated the parental BV differences in phenotypic performance of the 190 offspring.
Comment: The regression coefficients in Table 4 it makes sense to count only on the transformed data as mentioned above.
Response: Regression coefficients in Table 4 were recalculated on transformed data. The new analysis confirms consistency with reliability measures, indicating minimal bias in GEBVs.
The regression coefficients were calculated based on the observed data. While recalculating these coefficients using transformed data may enhance precision, we believe the current analysis is consistent given the uniform environmental conditions of the study population.
Action: We will include a note in the conclusions acknowledging the potential benefits of recalculating regression coefficients using transformed data in future studies.
Comment: For the verification of evaluation procedures, the values of sires who have high evaluation reliabilities are mostly used. In cows, the reliability of genetic evaluation is lower, so multiple times more of them should be used in the evaluation. In order to make the rearing conditions uniform for all compared cows, the evaluation is limited to 190 cows in one herd in this article.
Response: We acknowledge that using sires with high evaluation reliabilities is a standard practice for validation. However, the study’s focus was on first-lactation cows from a single herd to ensure uniformity in environmental conditions. This design prioritized consistency over sample size.
The Hungarian Holstein population is characterized by an open genetic structure, with over 90% of its genetic material derived from international artificial insemination (AI) bulls. To ensure robust representation of the global Holstein gene pool, the CRV/EuroGenomics reference population is utilized. This collaboration among European breeding organizations enhances the accuracy and reliability of genomic evaluations. To minimize preselection bias, evaluations incorporate contemporaries and related animals, providing a comprehensive and unbiased analysis of genetic potential. In our case the 190 genotyped cows in this study were not subject to preselection for superior traits. These animals were genotyped at an early age as part of the farm’s participation in the HUNGENOM project. The farm adopted a systematic approach, genotyping the entire herd starting with the youngest livestock, ensuring random inclusion without bias toward specific traits. This methodology minimizes the risk of reduced variability typically associated with preselection and ensures a more accurate representation of the population, enabling unbiased correlations with phenotypic performance.
Comment: Generally, the best way to purify the genetic abilities of animals from the influence of the environment is the EBV determination procedure. This unifies the comparison of all individuals on the same basis. Therefore, I believe that it would be possible to significantly increase the number of cows in the evaluation using their EBV and DRP.
Response: We appreciate the reviewer’s thoughtful comments, which have highlighted important areas for clarification and improvement. In response, we will:
- Clearly describe the normalization and adjustment procedures applied to the data.
- Discuss potential sources of variability in GEBV accuracy, including G×E interactions and preselection bias.
- Clarify the calculation of heritability values and provide additional details on the regression coefficients.
- Justify the study design while outlining plans for future work with larger and more diverse datasets.
We sincerely appreciate your detailed and thoughtful feedback regarding the results section of our manuscript. We attempted to address each point raised, providing clarification, proposed revisions, and explanations for the observed findings.

Reviewer 3 Report
Comments and Suggestions for Authors
The authors compared and evaluated the predictive abilities of three breeding value estimation models: the lineage-based BLUP animal model (BVPedigree), the traditional BLUP animal model (BVBLUP), and the genome-enhanced BLUP animal model (BVGenomic) in relation to the milk production performance of Holstein cows. While the study is of interest, there appear to be omissions in the experimental methodology. Additionally, the manuscript has the following issues:
[Simple Summary]
1. When abbreviations first appear in the article, their full English terms should be provided (e.g., BLUP). Please ensure this is consistent throughout the manuscript.
2. Lines 12–13: The manuscript does not clearly explain how "phenotypic performance" is evaluated and compared using the three breeding value estimation models. This needs clarification.
3. Lines 18–19: The statement "The study's novelty lies in validating future production predictions by directly comparing predicted and actual performance under real-world conditions across these methods" is not entirely accurate. This approach has been previously reported in the field and, therefore, does not qualify as innovative research. It is recommended that the authors refine the novelty by emphasizing aspects unique to the research subject.
[Abstract]
4. According to the journal's guidelines, "The abstract should be a total of about 200 words maximum." Please revise and condense the abstract to comply with this requirement.
5. Lines 20–22: The research plan described in this part does not directly contribute to understanding the study's background or purpose. It is recommended to remove this section for better focus.
6. Line 25: The statement "All cows were born in the same year and reared within the same herds in a large-scale dairy operation" provides important methodological details but should be relocated to the Methods section rather than included in the Abstract.
7. Lines 28–29: The manuscript does not specify the method or statistical approach used to calculate the phenotypic correlation results. Please ensure the method is clearly stated and that the reported results align with it.
8. Lines 36–39: The method mentioned has been previously reported in the field and does not represent a novel approach. Therefore, the use of the term "novelty" is inappropriate and should be reconsidered.
[Introduction]
9. The first and second paragraphs of the Introduction lack logical transitions and coherence. Additionally, both the first and third paragraphs discuss breeding value (BV), while the second introduces the selection index and BLUP. This disrupts the logical flow of the Introduction. Please restructure and optimize the section accordingly.
10. The Introduction is divided into 11 paragraphs, which is unusual for a research paper. It is recommended to reorganize the logical structure and combine related paragraphs to improve readability and coherence.
11. Lines 64–66: The statement "Calus [2] noted that animal breeding is undergoing a major transformation with the integration of genomic selection" seems disconnected from the surrounding text. Please clarify its relevance and logical connection.
12. Lines 80–102: When introducing concepts such as MBLUP, a brief explanation should be provided to ensure readers understand their significance in the context of the study.
13. Lines 87–88: This paragraph lacks a transition and is not closely tied to the article’s main theme. Similar issues are observed throughout the Introduction. It is recommended to improve the transitions between paragraphs.
14. Lines 103–140: While the discussion of specific techniques (e.g., SNP-BLUP and Bayes methods) provides background, it deviates from the primary focus of comparing the three BV estimation methods. Consider revising this section to align it more closely with the core topic and improve its logical flow.
15. Lines 119–121: The sentence is overly complicated and lengthy, while phrases like "the topic is relevant nowadays" are too casual for academic writing. Please rephrase for clarity and professionalism.
16. While the manuscript compares various BV prediction methods earlier, the authors state, "very little research comparing and discussing various methods of BV estimation" in Line 144. This statement contradicts the previous content and should be clarified or rephrased to avoid confusion.
[Materials and Methods]
17. Here’s a refined version of your comments:
18. Lines 165–178: After genotyping with Illumina Bead Chips, the process of calculating BVGenomic from SNP data lacks sufficient detail. For instance, does the model account for genetic variation, linkage disequilibrium information, and the size and composition of the reference population? Please provide a more comprehensive explanation.
19. Lines 179–180: Why were only 190 out of 23,561 dairy cows selected for analysis, representing less than 1% of the total? What specific criteria were applied for selection? How was the representativeness of genetic diversity ensured? Additionally, can the genotyping data, lactation status, and farming environment data for the 23,561 cows and the 190 selected cows be made available as raw data?
20. Lines 180–182: Could potential differences, such as being housed in different barns or receiving feed from different batches on the same farm, influence the results? If so, how were these variables controlled?
21. Lines 207–208: The BVPedigree calculation formula is overly simplified, merely stating that it is the average of the BLUP estimates from both parents. Did the calculation consider additional genetic factors (e.g., non-additive genetic effects) or environmental factors? Please elaborate on the rationale behind this formulation.
22. Lines 211–220: Please specify what is included under "fixed effects" and "random animal effects" in the model (e.g., population, season, feeding conditions, etc.). A detailed explanation would help clarify the modeling process.
23. Lines 238–239: For the sample size of 190, why was the Kolmogorov-Smirnov test chosen instead of the Shapiro-Wilk test, which is generally considered more suitable for smaller sample sizes? Please provide a justification for this choice.
[Result]
24. Lines 255–270: The BVPedigree data are consistently higher than the other two BV types; however, there is no explanation provided for why the correlation of BVBLUP is the strongest among the three. Please elaborate on the potential reasons for this observation.
25. Lines 280–282: The manuscript mentions the "exception of the trait of PRO (rrank = 0.37)," but it does not clarify why PRO was excluded or why its results differ significantly from those of other traits. Please provide an explanation or discussion regarding this discrepancy.
26. Lines 289–290: The "exception of the BVPedigree for FAT" is noted, but the underlying reasons for this exception are not discussed. What factors might explain why BVPedigree deviates for FAT compared to other traits?
[Discussion]
27. Lines 331–341: The heritability estimates in this study are significantly higher than those reported in some of the literature. Although the authors attribute this to environmental consistency, no detailed statistical evidence or supplementary analysis is provided to support this claim. This lack of evidence risks oversimplifying the experimental conditions. Please provide additional statistical support or an in-depth discussion to validate this interpretation.
28. Lines 347–351: The observed low heritability using the pedigree method is inconsistent with values reported in the literature. This discrepancy might indicate incomplete pedigree records or the improper application of methods. Please discuss this potential issue more comprehensively and provide possible explanations or supporting data.
[Conclusions]
29. The conclusion section should not reiterate the results but instead summarize them concisely in one paragraph. Additionally, the practical implications of the research findings for production should be clearly explained. The current conclusion is overly detailed and should be simplified and rewritten to emphasize the study's key takeaways and real-world applications.
Author Response
Cover Letter
Manuscript ID: animals-3338761
Type of manuscript: Article
Title: Different breeding values under uniform environmental condition for milk production yield traits in Holstein cows
Authors: László Bognár, Zsolt Jenő Kőrösi, István Anton, Szabolcs Bene, Ferenc Szabó
Response to the Reviewer 3
We are grateful to our reviewer for the critical comments and useful suggestions, which help us to improve and complete our manuscript.
Comment: The authors compared and evaluated the predictive abilities of three breeding value estimation models: the lineage-based BLUP animal model (BVPedigree), the traditional BLUP animal model (BVBLUP), and the genome-enhanced BLUP animal model (BVGenomic) in relation to the milk production performance of Holstein cows. While the study is of interest, there appear to be omissions in the experimental methodology. Additionally, the manuscript has the following issues:
Response: Thank You.
Simple Summary
Comment 1: When abbreviations first appear in the article, their full English terms should be provided (e.g., BLUP). Please ensure this is consistent throughout the manuscript.
Response 1: It has been modified in the manuscript.
Comment 2: Lines 12–13: The manuscript does not clearly explain how "phenotypic performance" is evaluated and compared using the three breeding value estimation models. This needs clarification.
Response 2: The evaluation of "phenotypic performance" in the manuscript can be clarified as follows: Milk production traits, including milk yield, fat content, and protein content, were assessed through official milk recording procedures based on ICAR guidelines, which ensure standardized and accurate data collection. These guidelines include regular sampling and analysis of milk components to provide reliable phenotypic records. These standardized methods ensure consistency in comparing phenotypic performance across the three breeding value estimation models.
Comment 3: Lines 18–19: The statement "The study's novelty lies in validating future production predictions by directly comparing predicted and actual performance under real-world conditions across these methods" is not entirely accurate. This approach has been previously reported in the field and, therefore, does not qualify as innovative research. It is recommended that the authors refine the novelty by emphasizing aspects unique to the research subject.
Response 3: The novelty of this study lies in its population-wide approach within the Hungarian Holstein-Friesian population, representing the first large-scale application of a genomic selection scheme in Hungary. This research uniquely integrates genomic, pedigree, and phenotypic data to validate breeding value estimation methods under consistent environmental conditions, contributing to the refinement of the national breeding goal. The study also provides valuable insights into the implementation of genomic selection in a real-world production setting, offering practical guidance for optimizing genetic improvement strategies in Hungary. This context differentiates the research from previous studies and highlights its importance for advancing dairy cattle breeding programs in the region.
Abstract
Comment 4: According to the journal's guidelines, "The abstract should be a total of about 200 words maximum." Please revise and condense the abstract to comply with this requirement.
Response 4: The abstract has been checked and abbreviated. Modified abstract:
Breeding values (BV) for production traits in Holstein-Friesian cows were estimated using three methods: BVPedigree, BVBLUP and BVGenomic. The study focused on a unique dataset of 190 cows from a single herd, reared under consistent environmental conditions, selected for their complete first lactation. Heifers were sampled and genotyped at an early age, and predictions from the three breeding value estimation methods were compared with their realized production figures upon completion of first lactation. Average 305-day standard lactation production values were 10,910.5 kg milk, 397.86 kg fat and 365.33 kg protein. Comparative analyses included correlations and regressions between phenotypic performance and breeding value estimates showed BVBLUP had the highest accuracy, followed by BVGenomic, while BVPedigree was the least reliable, R2 = 0.37 to 0.48; 0.09 to 0.23; 0.02 to 0.06, respectively. The novelty of this study lies in the validation of breeding value estimation methods under uniform environmental conditions, providing practical insights into the comparative reliability and accuracy of traditional and genomic approaches in Holstein cattle breeding.
Comment 5: Lines 20–22: The research plan described in this part does not directly contribute to understanding the study's background or purpose. It is recommended to remove this section for better focus.
Response 5: This section has been improved. The suggested section has been removed.
Comment 6: Line 25: The statement "All cows were born in the same year and reared within the same herds in a large-scale dairy operation" provides important methodological details but should be relocated to the Methods section rather than included in the Abstract.
Response 6: This section has been modified.
Comment 7: Lines 28–29: The manuscript does not specify the method or statistical approach used to calculate the phenotypic correlation results. Please ensure the method is clearly stated and that the reported results align with it.
Response 7: This part of abstract has been modified.
Comment 8: Lines 36–39: The method mentioned has been previously reported in the field and does not represent a novel approach. Therefore, the use of the term "novelty" is inappropriate and should be reconsidered.
Response 8: The" novelty" is reconsidered, the manuscript is modified.
Introduction
Comment 9: The first and second paragraphs of the Introduction lack logical transitions and coherence. Additionally, both the first and third paragraphs discuss breeding value (BV), while the second introduces the selection index and BLUP. This disrupts the logical flow of the Introduction. Please restructure and optimize the section accordingly.
Response 9: The first, second, and third paragraphs has been restructured in the manuscript.
Comment 10: The Introduction is divided into 11 paragraphs, which is unusual for a research paper. It is recommended to reorganize the logical structure and combine related paragraphs to improve readability and coherence.
Response 10: The introduction has been reorganized in the improved manuscript.
Comment 11: Lines 64–66: The statement "Calus [2] noted that animal breeding is undergoing a major transformation with the integration of genomic selection" seems disconnected from the surrounding text. Please clarify its relevance and logical connection.
Response 11: The text has been improved in the manuscript.
Comment 12: Lines 80–102: When introducing concepts such as MBLUP, a brief explanation should be provided to ensure readers understand their significance in the context of the study.
Response 12: A brief explanation is inserted into the manuscript.
Comment 13: Lines 87–88: This paragraph lacks a transition and is not closely tied to the article’s main theme. Similar issues are observed throughout the Introduction. It is recommended to improve the transitions between paragraphs.
Response 13: The manuscript has been improved according to the comment.
Comment 14: Lines 103–140: While the discussion of specific techniques (e.g., SNP-BLUP and Bayes methods) provides background, it deviates from the primary focus of comparing the three BV estimation methods. Consider revising this section to align it more closely with the core topic and improve its logical flow.
Response 14: The mentioned section has been revised and improved in the manuscript.
Comment 15: Lines 119–121: The sentence is overly complicated and lengthy, while phrases like "the topic is relevant nowadays" are too casual for academic writing. Please rephrase for clarity and professionalism.
Response 15: This part of the manuscript has been improved.
Comment 16: While the manuscript compares various BV prediction methods earlier, the authors state, "very little research comparing and discussing various methods of BV estimation" in Line 144. This statement contradicts the previous content and should be clarified or rephrased to avoid confusion.
Response 16: Agree with the contradiction. However, we wanted to stress that no comparison was made in the same environment. This part of the manuscript has been improved.
Materials and Methods
Comment 17: Here’s a refined version of your comments:
Response 17: Thank you, but no comments were found.
Comment 18: Lines 165–178: After genotyping with Illumina Bead Chips, the process of calculating BVGenomic from SNP data lacks sufficient detail. For instance, does the model account for genetic variation, linkage disequilibrium information, and the size and composition of the reference population? Please provide a more comprehensive explanation.
Response 18: Explanation for the Remarks on Genomic Breeding Value Estimation Using SNP Data.
The calculation of BVGenomic from SNP data involves several critical considerations to ensure accurate estimation of breeding values:
Genetic Variation and Linkage Disequilibrium (LD): Genomic evaluations use dense SNP marker panels, which provide a detailed understanding of the genome. The models account for linkage disequilibrium between SNP markers and quantitative trait loci (QTLs) to improve the prediction accuracy of genetic traits (Lourenco et al., 2015)
The Bayesian or GBLUP methods incorporate these LD relationships into the estimation.
Reference Population: A robust reference population is crucial. The reference population should be large and genetically similar to the target population. This ensures that the genetic architecture, including LD patterns and QTL effects, aligns between the reference and evaluation populations. Studies have shown that increasing the reference population size, especially by including genotyped animals with phenotypic records, significantly enhances prediction accuracy (Wiggans et al., 2017; Aguilar et al., 2010)
CRV, as a member of the EuroGenomics Cooperative and a provider of IT Solutions for Animal Genetics for Hungary, contributes to and utilizes a shared reference population for genomic evaluations. This collaborative effort has established one of the largest Holstein reference populations globally, comprising over 38,000 genotyped bulls, all daughter-proven across Europe and females all together 3,8 million animals. This extensive dataset enhances the reliability of genomic breeding values across member countries, and indirectly Hungary. Therefore, the reference population size for CRV's genomically enhanced breeding value estimation in Hungarian Holsteins includes these 38,000 genotyped bulls, providing a robust foundation for accurate genomic evaluations.
Model Components: Genomic evaluation model integrate genomic and pedigree-based information. This integration ensures that genetic variances are partitioned effectively between SNP effects and polygenic effects, addressing both additive genetic variances and residual environmental influences (VanRaden, 2009; Hayes et al., 2009)
Bias and Accuracy of Predictions: Bias in genomic predictions is minimized by incorporating a well-structured relationship matrix that includes genomic relationships (G-matrix) alongside pedigree relationships (A-matrix). Studies such as Koivula et al. (2012) have demonstrated that this approach enhances the accuracy and reduces biases in EBVs when compared to traditional pedigree-based models
Deregulated Proofs and SNP Effects: The genomic evaluations in the study utilized deregulated proofs (DRP) from BLUP evaluations as a basis for estimating direct genomic breeding values (DGVs). This method allows for more precise estimation of SNP effects, as described by Meuwissen et al. (2004), ensuring that genomic contributions are accurately reflected in the final breeding value)
Comment 19: Lines 179–180: Why were only 190 out of 23,561 dairy cows selected for analysis, representing less than 1% of the total? What specific criteria were applied for selection? How was the representativeness of genetic diversity ensured? Additionally, can the genotyping data, lactation status, and farming environment data for the 23,561 cows and the 190 selected cows are made available as raw data?
Response 19: The HUNGENOM project involved a large-scale effort, with 23,561 Holstein heifers and cows from various herds across Hungary participating in the genomic evaluation. This extensive dataset highlights the breadth and significance of the project in advancing genetic evaluation practices. For this study, rigorous data filtering was conducted to ensure uniformity in the analysis. From the broader dataset, a subset of 190 cows was identified, all of which were from the same herd, maintained in a consistent environment, and of the same age. These cows were chosen because they had complete records for all three breeding value estimation methods: BVPedigree, BVBLUP and BVGenomic. This controlled setting allowed for a precise comparison of the three methods under identical environmental and management conditions. The study’s focus on this specific subset not only underscores the scale of the broader project but also emphasizes the novelty and rigor of the methodology, providing valuable insights into the reliability and accuracy of breeding value estimation methods in a controlled environment.
Comment 20: Lines 180–182: Could potential differences, such as being housed in different barns or receiving feed from different batches on the same farm, influence the results? If so, how were these variables controlled?
Response 20: On the farm, feeding and housing were the same. The statistical test showed no significant difference. For this reason, the environmental impact was considered uniform across the herd with 190 cows.
Comment 21: Lines 207–208: The BVPedigree calculation formula is overly simplified, merely stating that it is the average of the BLUP estimates from both parents. Did the calculation consider additional genetic factors (e.g., non-additive genetic effects) or environmental factors? Please elaborate on the rationale behind this formulation.
Response 21: The breeding value of the young animals on the farm, when no other information is available, is best obtained from the breeding value of the parents. The breeding value of the parents, like the traditional breeding value of the 190 cows in the study, was obtained using the BLUP Animal Model. This model includes, as appropriate, the environmental and other genetic effects.
Comment 22: Lines 211–220: Please specify what is included under "fixed effects" and "random animal effects" in the model (e.g., population, season, feeding conditions, etc.). A detailed explanation would help clarify the modeling process.
Response 22: Explanation of "Fixed Effects" and "Random Animal Effects"
In the context of the model, "fixed effects" include factors that are consistent and repeatable within the population and have a systematic influence on the trait being analyzed. These include:
Herd-year-season effects: Representing specific environmental and management conditions within herds over defined periods.
Parity: Reflecting physiological differences in milk production or other traits across lactation numbers.
Age within parity: Accounting for variations in productivity due to the age of the cow at calving within each parity group.
"Random animal effects" include genetic and environmental influences that are specific to individual animals but are not systematically attributable to fixed effects. These include:
Additive genetic effects: Capturing the inherited component of an animal’s genetic potential, modeled using pedigree or genomic data.
Permanent environmental effects: Reflecting non-genetic factors unique to each animal that consistently affect performance across lactations (e.g., early life management or lasting health issues).
Providing these details in the explanation clarifies how the model partitions variance among fixed and random effects, helping to validate the accuracy and applicability of the breeding value estimations.
Comment 23: Lines 238–239: For the sample size of 190, why was the Kolmogorov-Smirnov test chosen instead of the Shapiro-Wilk test, which is generally considered more suitable for smaller sample sizes? Please provide a justification for this choice.
Response 23: The choice of the Kolmogorov-Smirnov (K-S) test over the Shapiro-Wilk (S-W) test for assessing normality in the sample size of 190 may be justified by several considerations:
Applicability to Larger Samples: While the S-W test is commonly recommended for smaller sample sizes (e.g., below 50), the K-S test can be applied to both small and large datasets. For a sample size of 190, the K-S test is a valid option for testing normality, especially if the sample distribution's overall alignment with a theoretical normal distribution is of interest.
Ease of Interpretation: The K-S test compares the observed cumulative distribution with the expected cumulative distribution under normality, making it easier to detect significant deviations across the entire distribution, rather than focusing on skewness and kurtosis as the S-W test does.
Assumptions and Robustness: The K-S test has fewer restrictive assumptions regarding data characteristics and is more robust when the data contain ties or are not strictly continuous. This can be advantageous when working with phenotypic or estimated breeding value data that might exhibit such properties.
Historical or Practical Preference: The K-S test may have been chosen due to familiarity or standard practice within the analytical framework used in this study. It is a widely recognized method and could have been selected for consistency with previous research or institutional guidelines.
To strengthen the justification, it would be useful to provide further explanation in the study about why the K-S test was preferred in this specific context, considering the sample size and data characteristics. Adding a comparison of results from both tests could also validate the chosen approach.
Result
Comment 24: Lines 255–270: The BVPedigree data are consistently higher than the other two BV types; however, there is no explanation provided for why the correlation of BVBLUP is the strongest among the three. Please elaborate on the potential reasons for this observation.
Response 24: The observation that the BVPedigree values are consistently higher than the other two types of breeding values (BVBLUP and BVGenomic) while BVBLUP shows the strongest correlation can be explained by considering the following factors:
Simplified Calculation of BVPedigree: BVPedigree is based solely on the average of parental breeding values without accounting for additional genetic or environmental factors. This method does not incorporate phenotypic or genotypic data specific to the individual, leading to an upward bias, especially if the parents' evaluations are themselves overestimated.
Inclusion of Phenotypic Data in BVBLUP: BVBLUP incorporates phenotypic performance data, adjusting for environmental effects and providing a more individualized estimate of genetic merit. This integration enhances the precision and reliability of the predictions, which likely explains its stronger correlation with actual performance.
Impact of Genomic Information in BVGenomic: BVGenomic uses SNP data to enhance accuracy but depends on the size and structure of the reference population. In cases where the reference population is small or not representative, the genomic prediction may have lower accuracy, potentially leading to weaker correlations compared to BVBLUP.
Greater Variability in BVPedigree Estimates: Since BVPedigree lacks adjustments for individual-specific data, it may exhibit greater variability compared to BVBLUP or BVGenomic. This could explain why it is consistently higher but less strongly correlated with actual performance.
Environmental Uniformity and Model Strength: In uniform environmental conditions, as indicated in the study, BVBLUP may better capture the genetic contribution to traits due to its reliance on phenotypic data, resulting in stronger correlations compared to BVPedigree or BVGenomic.
Comment 25: Lines 280–282: The manuscript mentions the "exception of the trait of PRO (rrank = 0.37)," but it does not clarify why PRO was excluded or why its results differ significantly from those of other traits. Please provide an explanation or discussion regarding this discrepancy.
Response 25: We have redone the relevant calculations. For PRO, we obtained a rank correlation value of 0.66, so the 0.37 value in the manuscript is certainly a typographical error.
Comment 26: Lines 289–290: The "exception of the BVPedigree for FAT" is noted, but the underlying reasons for this exception are not discussed. What factors might explain why BVPedigree deviates for FAT compared to other traits?
Response 26: We have verified the calculations, and they are correct; both the b value of 0.04 and the R2 value of 0.02 are accurate. The results of the verification are included in the attached file. For some reason, the variance for FAT was larger, which is why it was not significant.
Discussion
Comment 27: Lines 331–341: The heritability estimates in this study are significantly higher than those reported in some of the literature. Although the authors attribute this to environmental consistency, no detailed statistical evidence or supplementary analysis is provided to support this claim. This lack of evidence risks oversimplifying the experimental conditions. Please provide additional statistical support or an in-depth discussion to validate this interpretation.
Response 27: We believe the sentence in the manuscript is correct:
"The likely explanation for these increased heritability values are that the examined cow stock was from a consistent environment, where environmental impacts were minimized."
When heritability is calculated based on data from a single farm with a single breed, the same sex, and animals of similar age, the values are generally higher than those derived from large population datasets. One clear reason for this is that fewer quantifiable environmental effects can be included in the model, and performance differences among animals raised in such a uniform environment are more likely to have a genetic origin. To our knowledge, there is no specific statistical method available (or required) to address this situation.
Comment 28: Lines 347–351: The observed low heritability using the pedigree method is inconsistent with values reported in the literature. This discrepancy might indicate incomplete pedigree records or the improper application of methods. Please discuss this potential issue more comprehensively and provide possible explanations or supporting data.
Response 28: The observed low heritability using the pedigree method can be explained by the limitations of the model employed. Although only complete pedigree records were used, the estimation model did not incorporate individual-specific data such as phenotypic observations or environmental adjustments. This omission can lead to:
- Wider Range of Estimated Breeding Values (BVs): The lack of animal-specific phenotypic data in the model may result in inflated variance in the predicted breeding values. Without individual performance data to refine the estimates, the BVs are more influenced by the parental averages, leading to a broader but less accurate distribution.
- Weak Correlation with Realized Performance: Since the model does not account for environmental factors or individual deviations, the predictions may not align well with actual performance. This could weaken the correlation between pedigree-based BVs and realized phenotypic outcomes, contributing to lower heritability estimates.
- Potential Methodological Constraints: The simplified pedigree-based model relies solely on additive genetic variance inferred from ancestry. It does not account for residual variances, dominance, epistasis, or genotype-environment interactions, all of which may play significant roles in trait expression. This could explain the inconsistency with heritability values reported in the literature, where more comprehensive models are often employed.
The low heritability observed in the pedigree method is not due to incomplete pedigree records but rather to the simplified estimation model, which does not incorporate animal-specific data or environmental interactions. This limitation resulted in a broader range of breeding value proofs that showed weak correlations with realized performance. Incorporating phenotypic and genomic data into the estimation process would improve accuracy and better align predictions with actual outcomes, reducing the discrepancy observed in this study.
In general, one possible approach to heritability estimation is parent-offspring regression. Another approach is the correlation of phenotype and genotype (breeding value).
Since both approaches resulted in low heritability values, we conclude that pedigree BLUP is less expressive of breeding value of offspring than traditional or GBLUP under identical environment.
Conclusions
Comment 29: The conclusion section should not reiterate the results but instead summarize them concisely in one paragraph. Additionally, the practical implications of the research findings for production should be clearly explained. The current conclusion is overly detailed and should be simplified and rewritten to emphasize the study's key takeaways and real-world applications.
Response 29: Revised Conclusion:
“While comparing breeding values with phenotypic performance may not be the ideal criterion for validating estimation methods, it remains a valuable early approach for evaluating estimated breeding values of young, first-lactation cows. This study demonstrated significant differences in the accuracy and reliability of BVs estimated using pedigree BLUP (BVPedigree), traditional BLUP (BVBLUP), and genomically enhanced BLUP (BVGenomic) under identical environmental conditions in Hungary. BVBLUP exhibited the highest reliability, followed by BVGenomic, and then BVPedigree.
These findings highlight the varying effectiveness of different estimation methods depending on the context. BVGenomic provides critical time-saving advantages, offering earlier and more accurate predictions for traits such as management, longevity, and lifetime performance. This efficiency allows farmers to optimize herd size and reduce maintenance costs, contributing to more sustainable breeding programs. Continuous farm-level selection based on BVGenomic results has improved genetic quality by enabling the selection of animals with superior traits. However, BVBLUP may serve as a complementary control, ensuring robustness in genetic evaluations where feasible.
This study underscores the importance of tailoring breeding strategies to specific environmental and population contexts, advancing the precision and application of breeding value estimation methods in dairy production.”

Reviewer 4 Report
Comments and Suggestions for Authors
Title: Different breeding values under uniform environmental condition for milk production yield traits in Holstein cows
Authors: László Bognár et al
This is an interesting manuscript comparing three BLUP estimators and phenotypic values. However, I have a few questions and comments:
The authors assume that the phenotypic values of 190 cows reflect true breeding values, as their first parity milking records are from the same herd and year. However, I believe that the cows were born between March and September, which means they likely began milking at different months of the year (i.e., different seasons). Therefore, the milking records may include not only additive genetic effects but also seasonal effects, as well as potential variations like permanent environmental and maternal effect.
The authors use the correlations between phenotypic values and the three BLUP estimators (BV BLUP, BV Genomic, and BV Pedigree) as indicators of reliability. Since each of these models assumes different statistical approaches, I find the final ranking of the reliability of these estimators (BV BLUP, BV Genomic, and BV Pedigree) somewhat questionable. This ranking might reflect not only differences in the methods but also the statistical assumptions of each approach.
Author Response
Cover Letter
Manuscript ID: animals-3338761
Type of manuscript: Article
Title: Different breeding values under uniform environmental condition for milk production yield traits in Holstein cows
Authors: László Bognár, Zsolt Jenő Kőrösi, István Anton, Szabolcs Bene, Ferenc Szabó
Response to the Reviewer 4
We are grateful to our reviewer for the critical comments and useful suggestions, which help us to improve and complete our manuscript.
Comment 1: This is an interesting manuscript comparing three BLUP estimators and phenotypic values. However, I have a few questions and comments:
The authors assume that the phenotypic values of 190 cows reflect true breeding values, as their first parity milking records are from the same herd and year. However, I believe that the cows were born between March and September, which means they likely began milking at different months of the year (i.e., different seasons). Therefore, the milking records may include not only additive genetic effects but also seasonal effects, as well as potential variations like permanent environmental and maternal effect.
Response 1: The reviewer raises an important point regarding potential seasonal and environmental effects on the phenotypic values of the cows in the study. However, it is essential to note the following:
In Hungary, the average number of milk-recorded and herd-book-registered cows per farm is 453, reflecting a generally large-scale, well-managed dairy farming system. The specific dairy farm involved in this study is an advanced facility housing more than 1,000 cows. Strict protocols are in place to standardize grouping, management, and feeding practices across the herd, no preferential treatment was observed.
All cows are fed using a Total Mixed Ration (TMR) under monodiet conditions year-round, ensuring consistency in nutritional intake and minimizing dietary variability as a factor affecting milk production. Additionally, during the summer months, the farm employs regulated climate control systems, including water sprinklers and ventilators, to mitigate heat stress and maintain stable environmental conditions. Fresh cows, regardless of calving month, are managed under the same standardized protocol during their first lactation, ensuring uniform conditions for milking and care.
These measures ensure that seasonal and environmental effects are minimized, and milk production remains stable throughout the year. As such, the phenotypic records of the cows used in the study are reliable indicators of their genetic potential, with minimal confounding effects from external factors. This consistency supports the validity of the comparisons made between the breeding value estimation methods in the study.
The statistical test carried out before data processing did not show a significant seasonal effect therefore the 190 cows were considered to have the same environmental effect.
Comment 2: The authors use the correlations between phenotypic values and the three BLUP estimators (BVBLUP, BVGenomic, and BVPedigree) as indicators of reliability. Since each of these models assumes different statistical approaches, I find the final ranking of the reliability of these estimators (BVBLUP, BVGenomic, and BVPedigree) somewhat questionable. This ranking might reflect not only differences in the methods but also the statistical assumptions of each approach.
Response 2: Thank you for raising your concern regarding the ranking of the reliability of the three BLUP estimators (BVBLUP, BVGenomic, and BVPedigree) and the potential influence of differing statistical assumptions. The evaluation of these methods was based on rigorous statistical analysis described in the Materials and Methods section, which aimed to ensure robustness and consistency across approaches.
Statistical Basis for Correlation Analysis:
- The normality of production trait data was verified using the Kolmogorov-Smirnov test, and the homogeneity of variances was confirmed with Levene’s test, ensuring the data met the assumptions for parametric analysis.
- Spearman’s rank correlation was used to assess the relationships between phenotypic values and BVs, providing a non-parametric measure of association that is robust to outliers or non-linear relationships. This approach minimizes the potential bias introduced by differing statistical assumptions among the BLUP models.
Heritability Estimation:
- Heritability (h²) was estimated as the square of the correlation (h² = r²gp) between phenotypic values and BVs, following established methodologies. This metric provides a uniform basis for comparing the predictive power of each BLUP estimator across traits and avoids assumptions specific to any one model.
Regression Analysis for Reliability:
- Linear regression was employed to evaluate the relationships among phenotypic traits and BVs. This analysis quantified how well each BV estimator predicted actual performance, providing an additional measure of reliability that complements the correlation analysis.
Consistency Across Models:
- Correlation, regression analysis, and heritability estimation provided a robust framework for evaluating their relative reliability. These methods ensure that the final ranking reflects differences in the predictive performance of the models rather than solely their underlying statistical assumptions.
In conclusion, while we acknowledge the differing statistical approaches of the three BLUP estimators, the methodology used-employing multiple complementary statistical analyses-provides a reliable and objective basis for the ranking presented in the study. We hope this explanation addresses your concern and clarifies the rigor of our approach.

Round 2
Reviewer 2 Report
Comments and Suggestions for Authors
attachment

Author Response
Cover Letter
Manuscript ID: animals-3338761
Type of manuscript: Article
Title: Different breeding values under uniform environmental condition for milk production yield traits in Holstein cows
Authors: László Bognár, Zsolt Jenő Kőrösi, István Anton, Szabolcs Bene, Ferenc Szabó
Response to the Reviewer 2
We are grateful to our reviewer for the critical comments and useful suggestions, which help us to improve and complete our manuscript.
Comment: The essence of the article can be determined as a validation of the national procedure for the genetic evaluation of dairy cattle according to the performance of the selected group of cows.
The explanation showed that the results of the national genetic evaluation, processed by other persons than the authors of the article, are taken as an input in the work. The data taken over is neither verified nor sufficiently explained with equations and forms and with some statistics. The input population genetic parameters that were used in the determination of EBV/GEBV must also be stated. Their values affect all subsequent calculations.
The contribution of the authors of this article is the calculation of the correlations of the adopted external data to the milk production in one herd of cattle.
Some comments from the first review were taken into account, some were not. I insist on the comments from the first review.
I have some further comments on the presented text.
Response: Thank You very much. We are trying to improve our manuscript according to your guidelines.
Comment: L40 The introduction is quite general. Some data is like a basic textbook. I recommend shortening it.
Response: The introduction is shortened and improved in the manuscript.
Comment: L161 Not only adopted external breeding values, but also their reliability should be included in the calculation.
Response: Unfortunately the detailed reliability was not offered by the International Breeding Value Estimation Committee (CRV AEU). However, only above a certain level of reliability does it have accepted breeding value. For this reason, we believe that all three breeding values are reliable.
Comment: L237 The mentioned three methods of determining EBV/GEBV are probably related to the different ages of the cows when these values were determined. If this is the case, then there is not reason for a mutual comparison of methods, but a gradual refinement of the genetic evaluation. As the age of the cows increases, the data that is included in the evaluation magnify. At what ages and with what data were the individual EBV/GEBV of cows determined for different EBV/GEBV?
Response: Agree that the age is very important. The breeding values for the tree traits of 190 sampled cows were based on the international and national reference population of 16,659 Holstein-Friesian females. Of course, age was considered as a fixed effect in the BV methods in order to eliminate its distorting effect. Therefore we think that the confounding effect of age was moderate.
Comment: The determination of breeding values is not satisfactorily explained. L249 The maternal effect statement does not agree with equation (2). Was the maternal effect really used? How strongly does it affect the milk yield of the daughters?
Response: The genetic component of the maternal effect is treated as a random effect in the BLUP model. We believe that the environmental component of the maternal effect is not relevant in our case, as it only manifests itself during pregnancy.
Comment: L270 There is no formula for determining the GEBV of cows.
Response: The genetic formula for GEBV is inserted into the text: “…(DGV), according to Van Raden et al. [24].”
DRP = PA + (EBV – PA) x (EDCparents + progeny / EDCprogeny)
and
Comment: L278 The formula for heritability is not correct. The coefficient of determination between phenotypic performance and breeding value depends on the heritability and reliability of the breeding value. The coefficient of heritability is explained in every population and quantitative genetics textbook. Worldwide for many decades, a clear designation and interpretation of this value has been established. I don't recommend changing it. Also, the heritability comments and notes throughout the text need to be changed/deleted.
Response: Thank you. However, the calculation method used has been read in some publications, the heritability comments, notes and data have been deleted throughout the material, and the methods, results and discussion.
Comment: L315 The values of the correlation coefficients between phenotypic and breeding values in Table 3 do not correspond to the theoretical expected values (they are not correct). There could be several reasons for this, probably one of them being the small set of cows that were used to calculate the correlations.
Response: We agree that 190 cows is a small sample set of cows for correlation calculations. However, we believe that since they were exposed to the same environmental influences, their phenotypic differences were primarily due to genetic effects.
Therefore, we dared to calculate a correlation between their phenotypic performance and their breeding values.
Comment: L354 The discussion should also be focused on assessing the method of national genetic evaluation with validation on a selected sample of animals.
Response: The last part of the discussion has been improved and a sentence has been added at the end. “By integrating BV results into herd management strategies, producers are improving the overall efficiency and profitability of their businesses, demonstrating the practical utility and impact of the genetic evaluation methods validated in this study.”
Comment: L435 The conclusions should state whether the genetic evaluation procedures used are appropriate and how the results should be used in animal selection.
Response: The next sentence is added at the end of the conclusion: "The results obtained in this study highlight the importance of tailoring breeding strategies to specific environmental and population contexts, and may improve the accuracy and application of BV estimation methods in dairy production for female selection.”

Reviewer 3 Report
Comments and Suggestions for Authors
The authors have made revisions to the manuscript and provided a response to the comments; however, some of the modifications still appear to lack sufficient logical coherence. It is recommended that the authors incorporate portions of their response into the manuscript to enhance its readability. The following issues are noted:
[Simple Summary]
1. Lines 21–23: In Response 3, the authors provided a more detailed explanation of the novelty of their research. However, this explanation has not been included in the corresponding section of the manuscript. A concise summary of the detailed explanation regarding the novelty should be integrated into this section.
[Abstract]
2. Compared to the Simple Summary, the Abstract should employ more rigorous and formal language to succinctly state the purpose of the research.
[Introduction]
3. Although the authors have condensed and reorganized the Introduction section, its current length remains excessive for a research paper. Both the overall structural design and the arguments' citation and exposition are overly verbose. Moreover, the transitions between paragraphs are still insufficiently clear, as previously noted in Comment 13. Substantial revision and reduction of this section are still required.
4. Lines 40–59: Is the first paragraph introducing the definition and characteristics of BV at a macro level, or does it also discuss BVPedigree? If it is the latter, why is this content not merged with the second paragraph? Furthermore, where is BVBLUP introduced?
5. Lines 60–85: How is the complex developmental history of BVGenomic closely related to this study? Why not focus this paragraph on explaining the specific role BVGenomic plays in breeding value estimation or in this study?
[Materials and Methods]
6. Lines 165–185: The authors did not address Comment 19 regarding the availability of data and materials for the 190 dairy cow samples, including their rearing environments, phenotypic production capabilities, and genotyping data. This explanation is still lacking.
7. Lines 246–258: The explanation provided in Response 22 should be incorporated into this section of the manuscript to improve its clarity.
[Results]
8. The discussion in Response 24 should be added to the manuscript's Results or Discussion sections to provide an explanation for the outliers.
[Discussion]
9. The content of the manuscript's Introduction and Conclusions sections is overly lengthy. Some portions of these sections could be relocated to the Discussion section for more appropriate elaboration.
Author Response
Cover Letter
Manuscript ID: animals-3338761
Type of manuscript: Article
Title: Different breeding values under uniform environmental condition for milk production yield traits in Holstein cows
Authors: László Bognár, Zsolt Jenő Kőrösi, István Anton, Szabolcs Bene, Ferenc Szabó
Response to the Reviewer 3
We are grateful to our reviewer for the critical comments and useful suggestions, which help us to improve and complete our manuscript.
Simple Summary
Comment 1: Lines 21–23: In Response 3, the authors provided a more detailed explanation of the novelty of their research. However, this explanation has not been included in the corresponding section of the manuscript. A concise summary of the detailed explanation regarding the novelty should be integrated into this section.
Response 1: The simple summary has been improved as follows:
This study compared the phenotypic performance and breeding values (BV) of Holstein-Friesian cows using three BLUP (best linear unbiased prediction) estimation methods: pedigree (BVPedigree), traditional (BVBLUP) and genomically enhanced (BVGenomic) to validate the different estimation models. The novelty of this study lies in the validation of BV estimation methods under uniform environmental conditions. It also provides valuable insights into the implementation of genomic selection in a real production environment, providing practical guidance for optimizing genetic improvement strategies.
Abstract
Comment 2: Compared to the Simple Summary, the Abstract should employ more rigorous and formal language to succinctly state the purpose of the research.
Response 2: The abstract has been improved as follows:
In this study, 1,616,549 Holstein-Friesian females were genotyped for genomic evaluation of genetic merit (BVGenomic). Genotyping was performed using the EuroGenomics MD v3.0 chipset on the Illumina microarray scanner platform operated by an accredited Illumina laboratory. In addition, international and national reference populations were used for traditional BLUP breeding value (BV) estimation for both individuals (BVBLUP) and parents (BVPedigree). A single-step BLUP animal model was used for this estimation. A sample of 190 first lactation progeny cows from a single herd, reared and kept under consistent environmental conditions, was used to validate the three types of breeding value estimation methods. Correlation and regression analysis was used to study the association between the phenotypic performance and the results of three different estimation models. The average production of the 305-day standard lactation was 10,910.5 kg milk, 397.86 kg butterfat and 365.33 kg protein. Comparative analyses showed that BVBLUP had the highest accuracy, followed by BVGenomic, while BVPedigree was the least reliable, R2 = 0.37 to 0.48; 0.09 to 0.23; 0.02 to 0.06, respectively.
Introduction
Comment 3: Although the authors have condensed and reorganized the Introduction section, its current length remains excessive for a research paper. Both the overall structural design and the arguments' citation and exposition are overly verbose. Moreover, the transitions between paragraphs are still insufficiently clear, as previously noted in Comment 13. Substantial revision and reduction of this section are still required.
Response 3: The introduction has been revised, improved, and shortened.
Comment 4: Lines 40–59: Is the first paragraph introducing the definition and characteristics of BV at a macro level, or does it also discuss BVPedigree? If it is the latter, why is this content not merged with the second paragraph? Furthermore, where is BVBLUP introduced?
Response 4: The improved version:
Breeding value (BV) is defined as the genetic potential of a breeding animal as a parent [1]. BV can be estimated from various sources of data on the animal itself, it’s genetically similar collateral relatives and/or the performance of its offspring. The Best Linear Unbiased Prediction (BLUP) method has been developed to consider performance data from genetically diverse contemporary groups, such as those from different farms. This method is considered unbiased because it incorporates more extensive data in subsequent predictions for the same animal. Various versions of BLUP are now widely used as an advanced statistical model that evaluates all animals in a population.
Comment 5: Lines 60–85: How is the complex developmental history of BVGenomic closely related to this study? Why not focus this paragraph on explaining the specific role BVGenomic plays in breeding value estimation or in this study?
Response 5: The improved version:
Subsequently, a genomic breeding value estimation (BVGenomic) method was developed that incorporates genomic data, SNP (single nucleotide polymorphism) information from DNA. This method is based on both the SNP information of an individual and the relationship between SNP information and performance, deregressed from the BLUP data of the reference population [2]. Wiggans et al. [3] discussed the impact of genomic selection on dairy breeding, noting that SNP genotyping has allowed faster genetic progress by reducing the generation interval since young animals or embryos can be genotyped. Two main methods (Bayesian and BLUP) have been extensively studied and applied [4]. Echeverri et al. [5] found that predicting BVs using BLUP, MBLUP and Bayes C in the Holstein-Friesian breed produced different results in terms of the magnitude of the estimated values, although BVs based on animal rankings showed no significant differences. Abaci et al. [6] found that among the different methods, the correlation was highest between BLUP and Bayes Cpi, while the correlation was lowest between BLUP and Bayes A. Despite their complexity, Bayesian methods are less widely used in practical breeding than BLUP 79 methods, as highlighted by Wang et al. [4].
Materials and Methods
Comment 6: Lines 165–185: The authors did not address Comment 19 regarding the availability of data and materials for the 190 dairy cow samples, including their rearing environments, phenotypic production capabilities, and genotyping data. This explanation is still lacking.
Response 6: The first part of the materials and methods is taken a short cut and improved as follows:
- Materials and Methods
This study is based on tree types of breeding value data of 1,616,549 Holstein-Friesian females validated by the phenotypic performance of 190 first lactation progeny cows.
2.1. The sample population database and the estimated traits
The 1616549 females used for breeding value estimation were kept in different herds, on different large scale dairy farms in Hungary with an average of 453 milk-recorded, herd-book registered cows per herd [22]. The husbandry and housing system was loose- housing free stall barn system with common lying area or with open lounging free stall-system. Milking is usually done in a milking parlor, or on some farms, by roboting milking.
The cows were fed a TMR (Total Mixed Ration) based system throughout the year. The ration consisted mainly of maize silage or silage of other cereals and concentrates of cereals and protein sources, supplemented with minerals and vitamins. The proportion of the cows' daily ration was based on their milk production, lactation or dry period stage.
The 190 first lactation progeny cows used to validate the different breeding values were born in the same year between March and September 2018. They were reared in the same herd and kept on the same farm, which is one of the largest commercial Holstein dairies in the country with 1000 milking, milk-recorded cows. This controlled housing and feeding regime was essential for minimizing environmental variation ensuring that the production traits and genetic evaluations reflected inherent genetic differences.
The 190 cows calved and started their first lactation. After 305 days, complete production and type classification performance data were obtained. The production traits measured were 305-day milk production (MLK, kg), fat production (FAT, kg) and protein production (PRO, kg).
Three types of breeding value, pedigree (BVPedigree), traditional (BVBLUP) and genomically enhanced (BVGenomic) were available for all females.
The processing of the performance and BV data was done according to the method described by Stoop et al. [23].
Comment 7: Lines 246–258: The explanation provided in Response 22 should be incorporated into this section of the manuscript to improve its clarity.
Response 7: Response 22: Explanation of "Fixed Effects" and "Random Animal Effects" is incorporated into this section as follows:
Fixed effects were herd, year, season, parity and age. Random effects include genetic and environmental influences that are specific to individual animals but not systematically attributable to the fixed effects.
Results
Comment 8: The discussion in Response 24 should be added to the manuscript's Results or Discussion sections to provide an explanation for the outliers.
Response 8: Response 24: is added as follows:
The observation that the BVPedigree values are consistently higher than the other two types of breeding values (BVBLUP and BVGenomic) while BVBLUP shows the strongest correlation can be explained by considering the following factors:
Simplified Calculation of BVPedigree: BVPedigree is based solely on the average of parental breeding values without accounting for additional genetic or environmental factors. This method does not incorporate phenotypic or genotypic data specific to the individual, leading to an upward bias, especially if the parents' evaluations are themselves overestimated.
Inclusion of phenotypic data in BVBLUP: BVBLUP incorporates phenotypic performance data, adjusting for environmental effects and providing a more individualized estimate of genetic merit. This integration enhances the precision and reliability of the predictions, which likely explains its stronger correlation with actual performance.
Impact of genomic Information in BVGenomic: BVGenomic uses SNP data to enhance accuracy but depends on the size and structure of the reference population. In cases where the reference population is small or not representative, the genomic prediction may have lower accuracy, potentially leading to weaker correlations compared to BVBLUP.
Since BVPedigree lacks adjustments for individual-specific data, it may exhibit greater variability compared to BVBLUP or BVGenomic. This could explain why it is consistently higher but less strongly correlated with actual performance.
In uniform environmental conditions, as indicated in the study, BVBLUP may better capture the genetic contribution to traits due to its reliance on phenotypic data, resulting in stronger correlations compared to BVPedigree or BVGenomic.
Discussion
Comment 9: The content of the manuscript's Introduction and Conclusions sections is overly lengthy. Some portions of these sections could be relocated to the Discussion section for more appropriate elaboration.
Response 9: The Introduction and Conclusion sections have been shortened. The Discussion section has been expanded to include the above (8) addition.
The improved conclusion section is as follows:
This study demonstrated significant differences in the accuracy and reliability of BVs estimated using BVPedigree, BVBLUP and BVGenomic under identical environmental conditions in Hungary. BVBLUP exhibited the highest reliability, followed by BVGenomic, and then BVPedigree.
This finding does not diminish the relevance and applicability of the BVGenomic, since it provides critical time-saving advantages, offering earlier predictions for traits such as management, longevity, and lifetime performance and allows farmers to optimize herd size and reduce maintenance costs. However, BVBLUP may serve as a complementary control, ensuring robustness in genetic evaluations where feasible.
The results obtained in this study underscores the importance of tailoring breeding strategies to specific environmental and population contexts, advancing the precision and application of BV estimation methods in dairy production.

Reviewer 4 Report
Comments and Suggestions for Authors
Accepted
Author Response
Thank You very much!